
# Ownerless island and partial entanglement entropy in island phases

**Debarshi Basu[1][°], Jiong Lin[2,3][°], Yizhou Lu[4][°] and Qiang Wen[5][⋆]**

**1** Indian Institute of Technology, Kanpur 208016, India
**2** Interdisciplinary Center for Theoretical Study,
University of Science and Technology of China,
Hefei, Anhui 230026, China
**3** Peng Huanwu Center for Fundamental Theory, Hefei, Anhui 230026, China
**4** Department of Physics, Southern University of Science and Technology,
Shenzhen 518055, China
**5** Shing-Tung Yau Center and School of Physics, Southeast University, Nanjing 210096, China

⋆ wenqiang@seu.edu.cn

## Abstract

In the context of partial entanglement entropy (PEE), we study the entanglement structure of the island phases realized in several 2-dimensional holographic set-ups. From a pure quantum information perspective, the entanglement islands emerge from the self-encoding property of the system, which gives us new insights on the construction of the PEE and the physical interpretation of two-point functions of twist operators in island phases. With the contributions from the entanglement islands properly taken into account, we give a generalized prescription to construct PEE and balanced partial entanglement entropy (BPE). Here the ownerless island region, which lies inside the island $\mathrm{Is}(AB)$ of $A \cup B$ but outside $\mathrm{Is}(A) \cup \mathrm{Is}(B)$, plays a crucial role. Remarkably, we find that under different assignments for the ownerless island, we get different BPEs, which exactly correspond to different saddles of the entanglement wedge cross-section (EWCS) in the entanglement wedge of $A \cup B$. The assignments can be settled by choosing the one that minimizes the BPE. Furthermore, under this assignment we study the PEE and give a geometric picture for the PEE in holography, which is consistent with the geometric picture in the no-island phases.


doi:10.21468/SciPostPhys.15.6.227

## Contents

---

[°] These authors contributed equally to this work.

# 1 Introduction

In the past few decades there have been significant progress in our understanding of the quantum information aspects of black holes, which is quite useful for our understanding of the black hole information paradox [1]. In the context of the AdS/CFT correspondence [2], the von Neumann entropy $S_A$ for a region $A$ in the boundary CFT, was proposed to be dual to the

area of a minimal surface $\mathcal{E}_A$ in the bulk AdS geometry which is homologous to $A$ [3, 4],

$$S_A = \frac{\text{Area}(\mathcal{E}_A)}{4G_N} \,. \tag{1}$$

This relation between geometry and entanglement is the famous Ryu-Takayanagi (RT) formula. This formula is further refined to the quantum extremal surface (QES) formula [5, 6] with the first order quantum correction from the bulk fields included.

In [7, 8] the QES formula is applied to calculate the entanglement entropy of the Hawking radiation for an evaporating black hole after Page time. The black hole is in a 2-dimensional JT gravity, which is coupled to a non-gravitational CFT bath at the boundary. The new insight is that, after the Page time, a new QES behind the horizon becomes dominant, and the region behind the QES is included in the entanglement wedge of the Hawking radiation. These new insights are further refined towards the so called *island formula* [9–11]. The proposal of island phase and entanglement islands not only opens a new window for us to understand the quantum information aspects of black holes, but also introduces novel matter phases and entanglement structures in quantum systems.

The island formula claims that, when we calculate the entanglement entropy of a region $A$ in the non-gravitational bath, we should consider the possibility of including a region Is($A$) inside the gravitational region and calculate the entanglement entropy in the following way,

$$\text{Island formula } I: \quad S_A = \min \text{ext}_{\text{Is}(A)} \left\{ \frac{\text{Area}(\partial \, \text{Is}(A))}{4G_N} + S_{\text{bulk}}(A \cup \text{Is}(A)) \right\}. \tag{2}$$

More explicitly we consider all the possible island regions Is($A$) and choose the one that minimizes the sum inside the brackets of (2). If the entanglement entropy calculated by (2) is smaller than the one calculated in the usual way without islands, then the configuration enters the island phase and (2) is the correct way to calculate the entanglement entropy. After the Page time, the JT gravity coupled with the CFT bath enters the island phase, which produces exactly the decreasing part of the Page curve. Furthermore, the island formula has been derived via gravitational path integrals where wormholes are allowed to join the black holes in different copies of the configurations in the replica trick [10, 11]. Such a wormhole configuration turns out to be a new saddle when calculating the partition function in the replica manifold. Later it has been found that, the island phase can be realized in the AdS/BCFT setup [12] in a simple way, even without a black hole. See [13–48] for more references on recent developments on island formula in gravity and its applications.

Recently in [49], the island formula has been studied from a pure quantum information theoretic perspective. When a quantum system is constrained in such a way that its Hilbert space is substantially reduced and the state of a subset Is($A$) is completely encoded in the state of another subset $A$,[1] we call it a *self-encoded* system. Remarkably, in the self-encoded systems the entanglement entropy of $A$ should be calculated by a formula very similar to (2), which we have called the Island formula *II* [49],

$$\text{Island formula } II: \quad S_A = \frac{\text{Area}(\partial \, \text{Is}(A))}{4G_N} + \tilde{S}(A \cup \text{Is}(A)), \quad |A\rangle \Rightarrow |\text{Is}(A)\rangle \,. \tag{3}$$

In Island formula *II*, the first term proportional to the area of the boundary of Is($A$) arises if Is($A$) is settled in a gravitational background. The second term $\tilde{S}(A \cup \text{Is}(A))$ denotes the von

---

[1]Such confinements may be interpreted as projecting out certain states in the Hilbert space such that, for all remaining states in the reduced Hilbert space, the state of the subregion *Is($A$)* is determined by the state of the subregion *A* via a coding relation. The simplest example could be a two-spin system in which the four-dimensional Hilbert space $\mathcal{H} = \{|00\rangle, |01\rangle, |10\rangle, |11\rangle\}$ reduces to the two-dimensional space $\mathcal{H}_{reduced} = \{|00\rangle, |11\rangle\}$, such that the state of one spin in the reduced Hilbert space is completely determined by the state of the other spin. See [49] for more details.

Neumann entropy of the reduced density matrix is calculated by tracing out the degrees of freedom in the complement of $A \cup \mathrm{Is}(A)$ in a fixed geometric background. As was pointed out in [49], since the state of $\mathrm{Is}(A)$ is totally determined by the state of $A$, when we set boundary conditions for $A$ to compute the elements of the reduced density matrix $\rho_A$, we should simultaneously set boundary conditions for $\mathrm{Is}(A)$ following the coding relation between $A$ and $\mathrm{Is}(A)$. Consequently, there is no room to trace out the degrees of freedom in $\mathrm{Is}(A)$ because they are not independent with respect to $A$. More importantly, due to the self-encoding property, additional twist operators emerge at the boundary of $\mathrm{Is}(A)$, which means gravitation is not necessary for entanglement islands in this scenario. Nevertheless, due to the simplicity of the coding relation $|A\rangle \Rightarrow |\mathrm{Is}(A)\rangle$ under consideration, there is no optimization in the above Island formula $II$.

From a purely quantum information perspective, the self-encoding property may be the only plausible explanation for the emergence of entanglement islands. In [49], it was then boldly conjectured that the Island formula $I$ is indeed a special application of the Island formula $II$ in gravitational systems. This indicates that, gravitational systems with entanglement islands are self-encoded with a complicated coding relation,[2] such that the island formula $II$ becomes an optimization that exactly coincides with the island formula $I$. It was further pointed out in [49] that, the gravitational renormalization is possibly responsible to the vast reduction of the Hilbert space that result in the self-encoding property.

Inspired by this conjecture, a holographic $\mathrm{CFT}_2$ with no gravitation under a special Weyl transformation was proposed to sustain entanglement islands in [49]. This is the Set-up 1 where we analyze its island configurations and entanglement structure. The main reason we use this setup is that, in the effective theory description the two-point functions of twist operators which are not symmetric with respect to the $x = 0$ point is also well defined in this setup. For the readers who are not convinced by the application of the island formula to non-gravitational systems, we emphasize that the discussion in our paper is also valid when we couple the Weyl transformed part of the CFT to gravity (the gravitational Set-up 1). Also in the appendix, we introduce the Set-up 2, which is a generalized version of the AdS/BCFT setup where the non-symmetric two-point functions of twist operators can also be defined.

Given the setups, we focus on the so-called entanglement wedge cross-section (EWCS) in the gravitational dual of a boundary state in island phase and its dual quantum information quantity. The EWCS of the entanglement wedge of two non-overlapping regions $AB \equiv A \cup B$ is a natural measure for the mixed state correlation between $A$ and $B$. Since the measure for mixed state correlations is also not well studied in quantum information theory, the study of EWCS is also quite interesting from the quantum information perspective. In [50, 51], it was proposed that the quantum information quantity corresponding to the EWCS is the entanglement of purification, since it satisfies a similar set of inequalities as the EWCS. Since then, a series of quantum information quantities have been proposed to be the dual of the EWCS, which includes the entanglement negativity [52–54], the reflected entropy [55], the "odd entropy" [56], the "differential purification" [57], the entanglement distillation [58,59]. See [60–67] for more explorations along these lines. Nevertheless, most of these quantities are defined in terms of an optimization problem, which makes them extremely difficult to calculate and the evidence for their correspondence to the EWCS is not enough. The reflected entropy is defined as the entanglement entropy under a special canonical purification, hence calculable in general quantum systems. Moreover, the correspondence between the EWCS and the reflected entropy in island phases was explicitly studied in [68, 69]

In this work we will, in particular, study the balanced partial entanglement entropy (BPE) [70–72], which has also been proposed to be dual to the EWCS. For a purification $A_1 B_1 AB$ of the mixed state $\rho_{AB}$, the BPE is a special partial entanglement entropy (PEE) $s_{AA_1}(A)$ satisfy-

---

[2]Nevertheless, such a complicated coding relation is still not clear to us.

ing certain balanced conditions. The BPE is easy to calculate, as we have several powerful prescriptions to construct the PEE [73–75] in two dimensions. Moreover, unlike the reflected entropy and entanglement of purification which are defined on some special purifications, the BPE can be defined in generic purifications and is claimed to be purification independent. The purification independence and the correspondence to the EWCS for the BPE have been tested in global and Poincaré AdS$_3$ [71], holographic CFT$_2$ with an arbitrary Weyl transformation [71], holographic CFT$_2$ with gravitational anomalies [72] and BMS$_3$ symmetric field theories dual to 3-dimensional asymptotically flat spacetimes [71]. In particular, BPE can be regarded as a generalization of the reflected entropy, as BPE reduces to the reflected entropy for the particular case of canonical purification.

The main task of this paper is to study the BPE for island phase in 2-dimensional holographic theories and match it with the EWCS. This is a highly non-trivial task. Firstly, the phase structure of the EWCS is more complicated than the entanglement entropy (or RT surfaces), and we need to reproduce the phase structure from the BPE, which is purely evaluated from field theory side without any reference to the geometric picture. Secondly, in island phase when we calculate the entanglement entropy of a certain region $A$, it may involve other degrees of freedom outside $A$ there is an entanglement island Is($A$). This essentially change the way we calculate the entanglement entropy and PEE, hence we need to generalized the way we construct the PEE and BPE to the scenarios with entanglement islands. We find that the generalization involves the assignment of the contribution from the ownerless island regions. For two non-overlapping regions $A$ and $B$, when the entanglement island of $AB$ is larger than the union of the islands of $A$ and $B$, i.e.

$$\text{Is}(AB) \supset \text{Is}(A) \cup \text{Is}(B), \tag{4}$$

then the region Is($AB$)/(Is($A$) ∪ Is($B$)) inside Is($AB$) but outside Is($A$) ∪ Is($B$) is called the *ownerless island regions*. The ownerless island regions are closely related to the so-called reflected islands [68]. The key to calculate the BPE($A, B$) correctly, is to assign the contributions from the ownerless island regions to the right PEE. We will see that, different assignments for the ownerless island regions correspond to different balance point for the BPE, as well as different saddle point for the EWCS. We should choose the balance point that gives the minimal BPE or the EWCS saddle point with the minimal area. Eventually we get the matching between the BPE and EWCS.

This paper is organized as follows. In Sec.2, we will briefly introduce the partial entanglement entropy and the balanced partial entanglement entropy for usual quantum systems without islands. Then in section 3, we give the set-ups where island configurations are realized, and introduce a new concept of ownerless island regions and discuss how it changes the way we evaluate the PEE and BPE in island phase. After taking into account the contribution from the entanglement islands and ownerless islands, we generalize the way we compute the PEE and BPE to the scenarios with entanglement islands. In Sec.4 we give a classification for the EWCS in island phases in two dimensions. We also provide a naive calculation for BPE following the standard construction of the PEE and BPE in no-island phase, and find that it does not match with the EWCS. In Sec.5, with the generalized version of the ALC proposal (see (8)) and generalized balance requirements, we calculate the BPE for various configurations in island phase. We find that under different assignments of the ownerless island regions we get different BPEs, which correspond to different saddles of the EWCS. The minimal BPE matches exactly with the minimal EWCS. In section 6, under the assignments for the ownerless island that gives the minimal BPE, we evaluate the contributions $s_{AB}(A)$ and $s_{AB}(B)$ for various configurations and find consistency with the geometric picture. At last, in section 7 we give a summary of our results, discuss the physical significance of our results and provide an outlook for the future directions.

## 2 Brief introduction to PEE and BPE in non-island phase

### 2.1 Partial entanglement entropy

The *entanglement contour* is a concept in quantum information conjectured in [76], which is a function $s_A(\mathbf{x})$ that describes the contribution from each site $\mathbf{x}$ inside a subsystem $A$ to the entanglement entropy of $A$. This can be regarded as a density function of entanglement entropy inside $A$, that not only depends on the site $\mathbf{x}$ but also on the region $A$. By definition the entanglement contour function should satisfy

$$S_A = \int_A s_A(\mathbf{x})d\sigma_{\mathbf{x}}, \tag{5}$$

where $d\sigma_{\mathbf{x}}$ is the infinitesimal area element of $A$. Later a systematic study on the *partial entanglement entropy* (PEE) has been carried out in [73–75, 77]. The PEE is defined as the contribution from a subset $\alpha$ of $A$ to the entanglement entropy $S_A$, which can be expressed as

$$s_A(\alpha) = \int_\alpha s_A(\mathbf{x})d\sigma_{\mathbf{x}}. \tag{6}$$

The *entanglement contour* is a differential version of the PEE, and has been studied extensively in condensed matter theory to measure the spreading of entanglement under evolution [77–81]. In $AdS_3/CFT_2$, PEE correspond to bulk geodesic chords [73, 82] which is a finer correspondence between entanglement and geometry [73, 74]. More details on PEE, especially a first law-like version of the *entanglement contour* and its role in recently proposed island proposal can be found in [80, 83].

The expression $s_A(\alpha)$ displays the information about the contribution from the subregions, hence is called the contribution representation of the PEE. Later it was found that the PEE can be interpreted as an additive two-body correlation [75], and it is usually more convenient to express it in the following form

$$\mathcal{I}(\alpha, \bar{A}) \equiv s_A(\alpha), \tag{7}$$

where $\bar{A}$ is the complement of $A$ and $\bar{A} \cup A$ makes a pure state. We call the notation on the left hand side the *two-body-correlation representation* of the PEE, while the notation on the right hand side the *contribution representation*s of the PEE. These two representations are equivalent to each other in the usual quantum systems without islands.

The PEE should satisfy a set of physical requirements [75, 76] including those satisfied by the mutual information $I(A, B)$[3] and an additional one, the additivity property. For non-overlapping regions $A$, $B$ and $C$, the physical requirements for the PEE are classified in the following:

1. *Additivity:* $\mathcal{I}(A, B \cup C) = \mathcal{I}(A, B) + \mathcal{I}(A, C)$;

2. *Permutation symmetry:* $\mathcal{I}(A, B) = \mathcal{I}(B, A)$;

3. *Normalization:* $\mathcal{I}(A, \bar{A}) = S_A$;

4. *Positivity:* $\mathcal{I}(A, B) > 0$;

5. *Upper boundedness:* $\mathcal{I}(A, B) \leq \min\{S_A, S_B\}$;

6. $\mathcal{I}(A, B)$ *should be Invariant under local unitary transformations inside $A$ or $B$;*

---

[3]Note that, we should not mix between the the mutual information $I(A, B)$ and the PEE $\mathcal{I}(A, B)$.

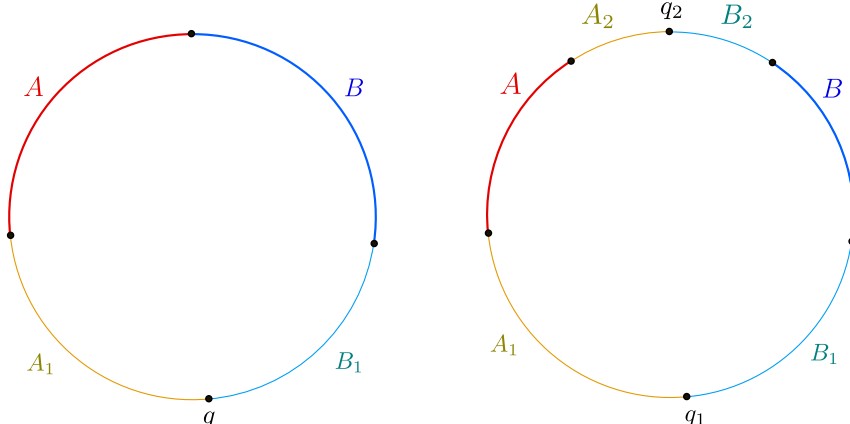

Figure 1: BPE for adjacent and disjoint intervals in CFT$_2$ vacuum. $q, q_1, q_2$ are balance points.

7. *Symmetry:* For any symmetry transformation $\mathcal{T}$ under which $\mathcal{T}A = A'$ and $\mathcal{T}B = B'$, we have $\mathcal{I}(A, B) = \mathcal{I}(A', B')$.

For more details about the well (or uniquely) defined scope of the PEE and the ways to construct the PEEs in different situations, the readers may consult [73–75, 77, 84–86]. These details are also summarized in the background introduction sections of [71,72]. Here we only introduce one particular proposal to construct the PEE in generic two-dimensional theories with all the degrees of freedom settled in a unique order (for example settled on a line or a circle), which we call the additive linear combination (ALC) proposal [73,75,77].

- **The ALC proposal**: Consider a region $A$ which is partitioned in the following way, $A = \alpha_L \cup \alpha \cup \alpha_R$, where $\alpha$ is some subregion inside $A$ and $\alpha_L$ ($\alpha_R$) denotes the regions left (right) to it. The *proposal* claims that:

$$s_A(\alpha) = \mathcal{I}(\alpha, \bar{A}) = \frac{1}{2}\left(S_{\alpha_L \cup \alpha} + S_{\alpha \cup \alpha_R} - S_{\alpha_L} - S_{\alpha_R}\right). \tag{8}$$

The Additivity and Permutation symmetry properties of the PEE indicate that, any PEE $\mathcal{I}(A, B)$ can be evaluated by the summation of all the two-point PEEs $\mathcal{I}(\mathbf{x}, \mathbf{y})$ with $\mathbf{x}$ and $\mathbf{y}$ located inside $A$ and $B$ respectively [75], i.e.

$$\mathcal{I}(A, B) = \int_A d\sigma_{\mathbf{x}} \int_B d\sigma_{\mathbf{y}} \, \mathcal{I}(\mathbf{x}, \mathbf{y}). \tag{9}$$

Note that the two-point PEE is an intrinsic entanglement structure of the system, in the sense that it is independent of the choice of the regions $A$ and $B$.

## 2.2 Balanced partial entanglement

Compared to the entanglement entropy, the entanglement contour or the PEE is a finer description for the entanglement structure of a quantum system. Then it is possible to extract other quantum information quantities from the PEE. In this paper, we focus on the so-called balanced partial entanglement entropy (BPE), which is a special PEE that satisfies certain balance conditions, and can be considered as a generalization of the reflected entropy in generic purifications of a mixed state. The BPE was proposed in [70] and is claimed to be dual to the EWCS. Furthermore in [71], it was proposed that the BPE captures exactly the reflected entropy in a mixed state and is purification independent. These proposals have passed various

tests in covariant scenarios [72], holographic $CFT_2$ with gravitational anomalies [72], $CFT_2$ with different purifications [71] and 3-dimensional flat holography [71,87].[4]

For a bipartite system $\mathcal{H}_A \otimes \mathcal{H}_B$ in a mixed state $\rho_{AB}$, one can introduce an auxiliary system $A_1 B_1$ to purify $AB$ such that the whole system $ABA_1 B_1$ is in a pure state $|\psi\rangle$, and $\text{Tr}_{A_1 B_1} |\psi\rangle \langle\psi| = \rho_{AB}$. The way we purify $\rho_{AB}$ is highly non-unique. The BPE between $A$ and $B$ is defined by

$$\text{BPE}(A:B) = \mathcal{I}(A, BB_1)|_{balanced} = \mathcal{I}(B, AA_1)|_{balanced} = s_{AA_1}(A)|_{balanced}, \tag{10}$$

where the subscript *balanced* means to satisfy the balance and minimal requirements, which are listed in the following:

1. **Balance requirement**: Among all possible configurations for the partition of $A_1 B_1$, we should choose the one satisfying the following condition

$$s_{AA_1}(A) = s_{BB_1}(B), \quad \text{or} \quad \mathcal{I}(A, B_1) = \mathcal{I}(A_1, B). \tag{11}$$

   When $A$ and $B$ are adjacent, (11) is enough to determine the partition of $A_1 B_1$, or equivalently, the balance point.

   However, when $A$ and $B$ are non-adjacent (see the right panel of Fig.1), the complement $\overline{AB}$ is disconnected and we need two partition points (see Fig.1) to divide $\overline{AB}$. In this case the balance requirements are generalized to two conditions

$$s_{AA_1 A_2}(A_1) = s_{BB_1 B_2}(B_1), \quad s_{AA_1 A_2}(A) = s_{BB_1 B_2}(B), \tag{12}$$

   or

$$\mathcal{I}(A_1, BB_2) = \mathcal{I}(B_1, AA_2), \quad \mathcal{I}(A, B_1 B_2) = \mathcal{I}(B, A_1 A_2). \tag{13}$$

   Since $S_{AA_1 A_2} = S_{BB_1 B_2}$, $s_{AA_1 A_2}(A_2) = s_{BB_1 B_2}(B_2)$ is automatically satisfied if the above two conditions are satisfied.

2. **Minimal requirement**: Usually the configurations for the partition of $A_1 B_1$ that satisfy the balance requirement are not unique. When there are multiple balance points, one should choose the one that minimizes $\text{BPE}(A:B)$. Later when we mention the balance requirements, the minimal requirement should be included.

Accordingly, for the configurations where $A$ and $B$ are non-adjacent, the definition of $\text{BPE}(A, B)$ generalizes to be

$$\text{BPE}(A:B) = \mathcal{I}(A, BB_1 B_2)|_{balanced} = s_{A_1 AA_2}(A)|_{balanced}. \tag{14}$$

Since at the balanced point $\mathcal{I}(A, B_1 B_2) = \mathcal{I}(A_1 A_2, B)$, the BPE can also be expressed as

$$\text{BPE}(A:B) = \mathcal{I}(A, B) + \frac{(\mathcal{I}(A, B_1 B_2) + \mathcal{I}(A_1 A_2, B))|_{balanced}}{2}. \tag{15}$$

**Minimal crossing PEE**: In the above expression for BPE, the first term is intrinsic, hence only the second term depend on the partition. In [71] it was observed that the summation $\mathcal{I}(A, B_1 B_2) + \mathcal{I}(A_1 A_2, B)$, called the *crossing PEE*, is purification independent and minimized at the balance point. This observation has been tested in both static [71] and covariant [72] configurations in $AdS_3/CFT_2$. In these cases the balance requirements can be replaced by an optimization problem, i.e. minimizing the crossing PEE. This is important because searching for the EWCS is also an optimization problem. It is interesting that, in $CFT_2$ when $A$ and $B$ are adjacent, this minimized crossing PEE is given by a universal constant which is the lower bound of a quantity termed the Markov gap [70,92,93].

---

[4]In 3-dimensional flat holography [88,89] the entanglement wedge and EWCS were studied in [90] based on the geometric picture of the holographic entanglement entropy [91] in flat holography .

# 3 Setups and the ownerless island regions

The island formula has been extensively studied in the models where a Jackiw-Teitelboim (JT) gravity coupled to a CFT$_2$ bath in flat background [5–8]. Combined with the braneworld holography [94–96], the entanglement islands also emerge in an effective 2d description of AdS/BCFT [12, 97–100]. Also, the PEE has been explored in this context; for example, the authors of [80, 83] have studied the entanglement contour for the Hawking radiation based on the straightforward application of the ALC proposal for PEE in island phase. In [101], the contribution from the island Is($A$) for certain region $A$ in the Hawking radiation was also discussed.

In this section, we will first introduce two setups with island phases. Then we study the PEE structure and their contribution to entanglement entropies in the presence of entanglement islands, which has not been thoroughly studies before. Furthermore, a new concept named the ownerless island regions are introduced, which are crucial for the evaluation of the BPE.

## 3.1 Set-up1: Holographic Weyl transformed CFT

The first setup with island phase is the holographic Weyl transformed CFT$_2$ proposed in [49]. Let us start from the vacuum state of a holographic CFT$_2$ on a Euclidean flat space with the metric $ds^2 = \frac{1}{\delta^2}\left(d\tau^2 + dx^2\right)$.[5] Here $\delta$ is an infinitesimal constant representing the UV cutoff of the boundary CFT. One may apply a Weyl transformation to the metric,

$$ds^2 = e^{2\varphi(x)}\left(\frac{d\tau^2 + dx^2}{\delta^2}\right), \tag{16}$$

which effectively changes the cutoff scale following $\delta \Rightarrow e^{-\varphi(x)}\delta$, where $\varphi(x)$ is usually negative. The entanglement entropy of a generic interval $A = [a, b]$ in the CFT after the Weyl transformation picks up additional contributions from the scalar field $\varphi(x)$ as follows [71, 102, 103]

$$S_A = \frac{c}{3}\log\left(\frac{b-a}{\delta}\right) + \frac{c}{6}\varphi(a) + \frac{c}{6}\varphi(b). \tag{17}$$

This formula can be achieved by performing the Weyl transformation on the two-point function of the twist operators.

Then we perform a UV cutoff dependent Weyl transformation for the $x < 0$ region such that the metric in this region is proportional to the AdS$_2$ metric,

$$\varphi(x) = \begin{cases} 0, & \text{if } x \geq 0, \\ -\log\left(\frac{2|x|}{\delta}\right) + \kappa, & \text{if } x < 0, \end{cases} \tag{18}$$

where $\kappa$ is an undetermined constant. After the Weyl transformation the metric at $x < 0$ becomes

$$ds^2 = \frac{e^{2\kappa}}{4}\left(\frac{d\tau^2 + dx^2}{x^2}\right), \qquad x < 0. \tag{19}$$

Such a special Weyl transformation changes the CFT essentially, and the cutoff scale at $x < 0$ is no longer related to $\delta$, rather it is characterized by the coordinate $x$. Note that, the scalar field (18) is non-smooth or even discontinuous at $x = 0$. Since the entropy (17) only depends on the scalar field at the endpoints, we think this is not a fatal problem as long as we do not talk about the intervals ending on the interface at $x = 0$. One can also redefine the scalar field

---

[5]Here the overall factor $\frac{1}{\delta^2}$ is inspired by AdS/CFT, where the metric is precisely the boundary metric of the dual AdS$_3$ geometry $ds^2 = \frac{\ell^2}{z^2}(-dt^2 + dx^2 + dz^2)$ with the radius coordinate settled to be $z = \delta$.

in the neighborhood of $x = 0$ to retain smoothness there. This will not affect our following discussions.

In [49], the subjection of the term $|\varphi(x)|$ in the entanglement entropy formula (17) was interpreted as putting a cutoff sphere in the AdS$_3$ background with radius $|\varphi(x)|$ and center at $(\delta, x)$. Assuming this interpretation, we put a cutoff sphere with radius $|\varphi(x)|$ for all the points with $x < 0$ for the specific Weyl transformation (18). The common tangent line of all these cutoff spheres in this case form a straight line in the *AdS* space (see Fig.2) which we call the cutoff brane. As stressed in [49], in this configuration the RT surfaces are allowed to be anchored on the cutoff brane in the sense that, the RT surfaces for symmetric intervals $[-a, a]$ are cut off there. This indicates that the cutoff brane plays a similar role as the end of world (EoW) brane in the AdS/BCFT setup [12] (see the appendix for a brief introduction). Furthermore, the parameter $\kappa$ plays a similar role as the tension of the EoW brane because the cutoff brane exactly settles at $\rho = \kappa$, where the coordinate $\rho$ is defined in the appendix of Ref. [49]. On the other hand, unlike the AdS/BCFT setup, here the degrees of freedom at $x < 0$ still settle at the asymptotic boundary rather than on the cutoff brane, and we have not assumed a gravitational theory on the cutoff brane. Nevertheless, later we will directly apply the island formula (2) on this system.[6] See [49] for more details about this configuration.

For readers who are not convinced by applying the island formula to non-gravitational theories, we can modify this setup accordingly. More explicitly we assume that the left-hand-side CFT in AdS$_2$ background is coupled to an induced gravity where the full gravity action is produced by integrating the matter CFT degrees of freedom. Here the AdS$_2$ background on the left hand side is also considered as a result of performing the Weyl transformation characterized by (18). This is just the setup introduced in the section 2 of [99], with the main difference being that the choice of the scalar field (18) in our paper includes the new parameter $\kappa$. In this setup the correlation functions of the twist operators are exactly the same as (17) in the holographic Weyl transformed CFT$_2$, and the holographic picture is also conjectured to be the AdS$_3$ with a EoW brane setted at $\rho = \kappa$ [49, 99]. Although the interpretation for the two setups could be different, the calculation for the entropy, PEE and BPE are exactly the same. We refer this modified setup the gravitational Set-up 1.

Before we go ahead, we also provide an alternative Set-up 2 in the appendix A. This setup is a generalized version of AdS/BCFT [9, 97–99, 104], where we add conformal matter to the EoW brane in the standard AdS/BCFT setup [12]. We will focus on the typical model called the defect extremal surface (DES) model proposed in [97], which treats the EoW brane as a defect and assumes a defect theory living the EoW brane. The 2-dimensional effective theory description for DES model is then described by a gravitational CFT$_2$ on the brane coupled to a bath CFT$_2$ with a transparent boundary condition, where the two-point functions of the twist operators for non-symmetric intervals can be defined. As in the previous setups, we get the two-point functions of twist operators under the same Weyl transformation. In this setup, the conformal matter on the brane in the DES model will introduce an additional defect term to the BPE or the reflected entropy, which should be understood as the contribution from the bulk entanglement entropy in the RT formula with quantum correction [105]. We will see

---

[6]If the RT surfaces can be cut off deep inside the bulk, then new minimal saddles will arise when we apply the replica trick in the AdS bulk following the Lewkowycz-Maldacena prescription [5]. For example, let us calculate the entanglement entropy for an interval $[a, \infty)$ with $a > 0$, the extremal surface with the minimal length will coincide with the RT surface of $[-a, a]$ which is cut off at the cutoff brane.. This is one of the main reasons for the authors of [49] to propose that, island formula can be applied in this holographic Weyl transformed CFT$_2$. The second reason comes from the existence of entanglement islands in non-gravitational systems which are self-encoded [49]. This opens a window for us to apply the island formula in this non-gravitational Weyl transformed CFT if the self-encoding property emerges due to the Weyl transformation, despite the fact that the previous arguments [10, 11] for entanglement islands via the replica wormholes relies on gravitation. The third reason is that, in this setup the island formula can give a smaller entanglement entropy in certain scenarios than the usual formula without islands, which we will see in the next subsection.

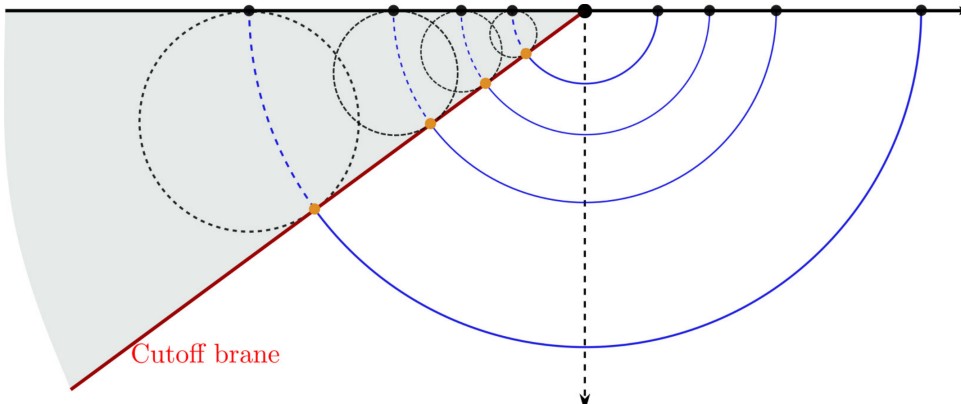

Cutoff brane

Figure 2: Figures extracted from [49], licenced under CC-BY 4.0. When we only consider symmetric intervals $[-a, a]$, the common tangent surface to all the cutoff spheres settled at $\rho = \kappa$, can be compared to the EoW brane ($\rho = \rho_0$) in the AdS/BCFT scenario. However, the RT surfaces for non-symmetric intervals will be cut off at certain cutoff sphere behind the cutoff brane, see Fig.19.

that, provided $\kappa = \rho_0$ (where $\rho_0$ is the tilt angle of the EoW brane in the DES model) the calculations for the BPE exactly match with the reflected entropy given in [106], which is the EWCS term plus an additional defect term that originates from the conformal matters on the brane in the DES model.

Later in this paper, for all the figures we will contract the region behind the cutoff brane (the gray region in Fig.2) to make them consistent with the widely used AdS/BCFT setup.

## 3.2 Islands in holographic Weyl transformed CFT

Let us apply the Island formula (2) to the (gravitational) Set-up 1. We consider $A$ to be the semi-infinite region $x > a$. The entanglement entropy calculated by the island formula is given by

$$S_{\mathcal{R}} = \min\left\{ \frac{c}{3} \log \frac{a + a'}{\delta} - \frac{c}{6} \log\left(\frac{2a'}{\delta}\right) + \frac{c}{6}\kappa \right\}, \tag{20}$$

where we take the region $I = (-\infty, -a']$ as a possible choice for Is($A$). One can easily check that, when $a = a'$ the above expression attains its minimal value:[7]

$$S_{\mathcal{R}} = \frac{c}{6} \log\left(\frac{2a}{\delta}\right) + \frac{c}{6}\kappa, \qquad a' = a, \tag{21}$$

which is always smaller than the entanglement entropy computed through the usual techniques. Similarly when we consider $A$ to be an interval $[a, b]$ inside the region $x > 0$ and $[-b', -a']$ is any possible choice of Is($A$), the island formula will give[8]

$$S_A = \min\left\{ \frac{c}{3} \log \frac{a + a'}{\delta} + \frac{c}{3} \log \frac{b + b'}{\delta} - \frac{c}{6} \log\left(\frac{2a'}{\delta}\right) - \frac{c}{6} \log\left(\frac{2b'}{\delta}\right) + \frac{c}{3}\kappa \right\}, \tag{22}$$

---

[7] Note that, in Set-up 1 the two terms in (21) are given by the Weyl transformed two-point function of twist operators for the interval $[-a, a]$. In the gravitational Set-up 1 the constant term $\frac{c}{6}\kappa$ is interpreted as the area term $\frac{Area(X)}{4G}$ since the gravity coupled to the left-hand-side CFT is induced by a simple partial reduction of the AdS$_3$ spacetime, see the appendix or [97,99] for more details.

[8] Here we need to assume that the entanglement entropy for two disjoint intervals in the holographic Weyl transformed CFT exhibit similar phase transitions as the RT formula [107,108], under certain sparseness conditions on the spectrum and OPE coefficients of bulk and boundary operators and large $c$ limit. We leave this for future investigation.

which has the saddle point

$$S_A = \frac{c}{6} \log\left(\frac{4ab}{\delta^2}\right) + \frac{c}{3}\kappa, \qquad a' = a, \quad b' = b. \tag{23}$$

As was found in [49], the above result is smaller than the usually computed entanglement entropy $S_A = \frac{c}{3} \log\left(\frac{b-a}{\delta}\right)$, when

$$a/b < r^* \equiv 1 - 2\sqrt{e^{2\kappa} + e^{4\kappa}} + 2e^{2\kappa}. \tag{24}$$

In other words, for any interval $[a, b]$ with $a/b < r^*$, the interval admits an island and the entanglement entropy should be calculated through the island formula.

In the context of AdS/BCFT, we can either consider the regions with no islands or regions $[a, b]$ ($b > a > 0$) with reflection symmetric islands $[-b, -a]$. In the configurations with islands, the entanglement entropies are calculated by the two-point functions of twist operators settled at reflection symmetric sites. These are also well-defined in the holographic Weyl transformed CFT$_2$, as the RT surfaces can also anchor at the common tangent vertically. Moreover, in holographic Weyl transformed CFT$_2$, the Weyl transformed two-point functions for twist operators can be generalized to the non-symmetric configurations. More explicitly, we can set the two twist operators at $-b$ and $a$ with $a, b > 0$ and $a \neq b$.

The physical interpretation for these two-point functions are not clear so far. It is quite tempting to interpret these two-point functions as the entanglement entropies for the intervals $[-b, a]$ with no reflection symmetry. Nevertheless, in the following, we argue that this interpretation is not correct, and we will provide a more suitable interpretation in Sec.3.4. For $b < a$, although the two-point function is still computable according to the island formula $II$, the region $(0, a]$ determines the state of its island region $[-a, 0)$. When we set boundary conditions for $(0, a]$, we simultaneously set boundary conditions on $[-a, 0)$, which means we can only trace the degrees of freedom outside $[-a, a]$, which is a larger region covering $[-b, a]$. This implies that the entanglement entropy for $[-b, a]$ with $b < a$ is not well defined. Similar problems arise for the region $(-\infty, -b] \cup [a, \infty)$ with $b > a$.

The above problem does not arise in the cases with $b > a$, where the region $[-b, 0)$ covers the island $[-a, 0)$. Nevertheless, in these cases the entanglement entropy for $[-b, a]$ is also not well defined, due to the self-encoding property of the system, which makes the situation different from the normal systems where the degrees of freedom at different sites are independent of each other. When we compute the reduced density matrix on $[-b, a]$, we set boundary conditions on $[-b, -a] \cup [-a, a]$ to compute the elements of the reduced density matrix. Even though, settling down the state for the region $[-b, -a)$ does not totally determine the state of any degrees of freedom outside $[-b, a]$, it would in some way confine the space of the sates for the complement $(-\infty, -b] \cup [a, \infty)$. Hence while tracing out the degrees of freedom outside $[-b, a]$, we should not go through all the states in the Hilbert space of the complement. Another evidence is that, the entanglement entropy for the complement region $(-\infty, -b] \cup [a, \infty)$ is not well-defined in the sense that $(-\infty, -b]$ is a subregion of the island of the region $[a, \infty)$.

## 3.3 The PEE structure of the island phase and the ownerless island

Previously we have introduced two representations for the PEE, i.e. the *contribution representation* and the *two-body-correlation representation*. The contribution representation needs the input of the region $A$ and its subset $\alpha$, while the two-body-correlation representation is an intrinsic structure of the state which does not rely on the choices of the regions and their subsets. As we have shown that, given a region $A$ and one of its subsets $\alpha$, we can generate all

the contributions $s_A(\alpha)$ from $\mathcal{I}(x, y)$ by integration (or summation). So we can claim that the two representations are equivalent to each other.

In island phase, the two-body-correlation structure is still an intrinsic structure of the state, while the contribution representation changes a lot in island phase. The reason is that, in island phase when we talk about certain region $A$, it also involves other degrees of freedom (which is in the island region Is($A$)) outside $A$ when it admits an island. In this case, not only the degrees of freedom inside $A$, but also those in its island contribute to $S_A$. The self-encoding property of the state indicates that the degrees of freedom at different sites are no longer independent of each other, and essentially change the way we evaluate the entanglement entropy of a region $A$, as well as the contribution $s_A(\alpha)$ from the subset $\alpha$. Later in this paper, our discussion will be conducted mainly in the two-body-correlation representation, and we will only refer to $\mathcal{I}(A, B)$ as a PEE.

More explicitly, let us consider a region $A$ which admits island. The entanglement entropy $S_A$ is calculated by the island formula $S_A = \tilde{S}_{A \cup \text{Is}(A)}$, which is the von Neumann entropy of the reduced density matrix $\tilde{\rho}_{A \cup \text{Is}(A)}$ computed by tracing out the degrees of freedom outside $A \cup \text{Is}(A)$. This strongly indicates that, we should collect all the two-point PEEs $\mathcal{I}(x, y)$ with $x$ and $y$ located in $A \cup \text{Is}(A)$ and its complement separately, i.e.

$$S_A = \tilde{S}_{A \cup \text{Is}(A)} = \int_{x \in A \cup \text{Is}(A)} dx \int_{y \in \overline{A \cup \text{Is}(A)}} dy\, \mathcal{I}(x, y)\,. \tag{25}$$

A direct consequence is that, the normalization requirement $S_A = s_A(A) = \mathcal{I}(A, \bar{A})$ in the non-island phase breaks down as the island also contributes to $S_A$.

- *In other words, the PEE between Is($A$) and $\overline{A \cup \text{Is}(A)}$ should contribute to $S_A$ since the state of Is($A$) is totally determined by $A$ and should be understood as a window through which $A$ can entangle with $\overline{A \cup \text{Is}(A)}$.*

One can also write the above equation as

$$S_A = \mathcal{I}(A, \overline{A \cup \text{Is}(A)}) + \mathcal{I}(\text{Is}(A), \overline{A \cup \text{Is}(A)})\,, \tag{26}$$

which is quite different from the normalization property $S_A = \mathcal{I}(A, \bar{A})$ in non-island phase. In the above formula (26), the region $A$ can be either connected or disconnected.[9]

- *We conclude that, in island phase the relation $s_A(\alpha) = \mathcal{I}(\alpha, \bar{A})$ between the two representations as well as the ALC proposal no longer holds. When we compute the contribution $s_A(\alpha)$ from the two-body-correlation $\mathcal{I}(x, y)$, we should take into account the island configuration carefully.*

Now we discuss the entanglement entropy for the union of two non-overlapping regions $AB \equiv A \cup B$ and the contribution $s_{AB}(A)$ and $s_{AB}(B)$ in terms of the PEEs in two-body-correlation representation. These configurations have more complicated structures in the presence of islands. In the following, we classify the island configurations of these configurations into three classes and explicitly discuss how the island structure affects the contributions.

**Class 1**

In the first class, the region $AB$, as well as $A$ and $B$, does not admit an island. In this case, since all the regions do not involve degrees of freedom outside $AB$, $s_{AB}(A)$ and $s_{AB}(B)$ can be

---

[9]See also [101] for an earlier discussion of the formula (26) when $A$ is a connected interval.

calculated by the ALC proposal for PEE in the non-island phase,

$$
\begin{aligned}
s_{AB}(A) &= \frac{1}{2}\left(S_{AB} + S_A - S_B\right) = \mathcal{I}(A, \overline{AB}), \\
s_{AB}(B) &= \frac{1}{2}\left(S_{AB} + S_B - S_A\right) = \mathcal{I}(B, \overline{AB}).
\end{aligned}
\tag{27}
$$

**Class 2**

In the second class, all the three regions $AB$, $A$ and $B$ admit islands and

$$
\mathrm{Is}(AB) = \mathrm{Is}(A) \cup \mathrm{Is}(B). \tag{28}
$$

Let us denote $C \equiv \overline{AB \cup \mathrm{Is}(AB)}$. In this case, the entanglement entropy $S_{AB} = \mathcal{I}(AB \cup \mathrm{Is}(AB), C)$ has contribution $\mathcal{I}(\mathrm{Is}(AB), C)$ from $\mathrm{Is}(AB)$. Since $\mathrm{Is}(AB)$ is just the union of $\mathrm{Is}(A)$ and $\mathrm{Is}(B)$, this island contribution can be decomposed into $\mathcal{I}(\mathrm{Is}(A), C) + \mathcal{I}(\mathrm{Is}(B), C)$, where the two terms should be assigned to $s_{AB}(A)$ and $s_{AB}(B)$ respectively. More explicitly we have

$$
s_{AB}(A) = \mathcal{I}(A, C) + \mathcal{I}(\mathrm{Is}(A), C), \tag{29}
$$
$$
s_{AB}(B) = \mathcal{I}(B, C) + \mathcal{I}(\mathrm{Is}(B), C). \tag{30}
$$

These PEEs can be calculated by writing them as a linear combination of $\tilde{S}_\alpha$. For example,

$$
\begin{aligned}
s_{AB}(A) &= \mathcal{I}(A\,\mathrm{Is}(A), C) = \frac{1}{2}\left(\tilde{S}_{AB\cup\mathrm{Is}(AB)} + \tilde{S}_{A\cup\mathrm{Is}(A)} - \tilde{S}_{B\cup\mathrm{Is}(B)}\right), \\
s_{AB}(B) &= \mathcal{I}(B\,\mathrm{Is}(B), C) = \frac{1}{2}\left(\tilde{S}_{AB\cup\mathrm{Is}(AB)} + \tilde{S}_{B\cup\mathrm{Is}(B)} - \tilde{S}_{A\cup\mathrm{Is}(A)}\right).
\end{aligned}
\tag{31}
$$

According to the formula (26), it is evident that the contribution $s_{AB}(A)$ and $s_{AB}(B)$ can still be written as the linear combination of entanglement entropies (including island contribution) following the ALC proposal.

**Class 3**

In the third class, the region $AB$ admits an island $\mathrm{Is}(AB)$, but $\mathrm{Is}(A) \cup \mathrm{Is}(B)$ does not cover the full $\mathrm{Is}(AB)$, i.e.

$$
\mathrm{Is}(AB) \supset (\mathrm{Is}(A) \cup \mathrm{Is}(B)). \tag{32}
$$

Here we do not require the two subregions $A$ and $B$ to admit individual islands. If $A$ (or $B$) does not admit an island, then $\mathrm{Is}(A) = \emptyset$ (or $\mathrm{Is}(B) = \emptyset$) in (32). In this case, there are degrees of freedom that belong to $\mathrm{Is}(AB)$ but outside $\mathrm{Is}(A) \cup \mathrm{Is}(B)$. We call these degrees of freedom the ownerless island region and denote them as $\mathrm{Io}(AB)$, i.e.

$$
\textit{Ownerless island region}: \ \mathrm{Io}(AB) = \mathrm{Is}(AB)/(\mathrm{Is}(A) \cup \mathrm{Is}(B)). \tag{33}
$$

It is also possible that neither $A$ nor $B$ admits islands and hence the total island $\mathrm{Is}(AB)$ is ownerless. Let us again denote the complement of $AB \cup \mathrm{Is}(AB)$ as

$$
C \equiv \overline{AB \cup \mathrm{Is}(AB)}. \tag{34}
$$

It is clear that the ownerless island region contributes to the entanglement entropy $S_{AB}$, since

$$
S_{AB} = \mathcal{I}(AB \cup \mathrm{Is}(AB), C) = \mathcal{I}(\mathrm{Io}(AB), C) + \mathcal{I}(AB \cup \mathrm{Is}(A) \cup \mathrm{Is}(B), C). \tag{35}
$$

However, it is not clear whether we should assign this contribution $\mathcal{I}(\mathrm{Io}(AB), C)$ to $s_{AB}(A)$, $s_{AB}(B)$ or to both of them.

The assignment for the contribution from the ownerless island region has not been discussed before. Let us divide $\mathrm{Io}(AB)$ into two parts

$$\mathrm{Io}(AB) = \mathrm{Io}(A) \cup \mathrm{Io}(B), \tag{36}$$

where $\mathrm{Io}(A)$ is assumed to contribute to $s_{AB}(A)$, while $\mathrm{Io}(B)$ is assumed to contribute to $s_{AB}(B)$, i.e.

$$s_{AB}(A) = \mathcal{I}(A \cup \mathrm{Is}(A), C) + \mathcal{I}(\mathrm{Io}(A), C), \tag{37}$$

$$s_{AB}(B) = \mathcal{I}(B \cup \mathrm{Is}(B), C) + \mathcal{I}(\mathrm{Io}(B), C). \tag{38}$$

One can check that, if we naively apply the ALC proposal, then the contribution from the ownerless island regions will be missing which results in a wrong answer. So in these cases the ALC proposal no longer holds and we should treat the ownerless island regions carefully.

Later we will see that different assignments will give us different BPEs which correspond to different saddles of the EWCS. Furthermore, when we have multiple balance points, we should choose the one that gives the minimal BPE, which helps us further determine the decomposition of the ownerless islands. Eventually we will see that the region $\mathrm{Io}(A) \cup \mathrm{Is}(A)$ (or $\mathrm{Io}(B) \cup \mathrm{Is}(B)$) will coincide with the so-called reflected entropy island of $A$ (or $B$) defined in [68]. For simplicity we define

$$\mathrm{Ir}(A) \equiv \mathrm{Is}(A) \cup \mathrm{Io}(A), \qquad \mathrm{Ir}(B) \equiv \mathrm{Is}(B) \cup \mathrm{Io}(B), \tag{39}$$

which can be understood as a generalized version of islands when calculating contributions from subsets with the ownerless island regions taken into account. The contributions in (37) then can be expressed as

$$s_{AB}(A) = \mathcal{I}(A \cup \mathrm{Ir}(A), C), \qquad s_{AB}(B) = \mathcal{I}(B \cup \mathrm{Ir}(B), C). \tag{40}$$

### 3.4 The generalized ALC proposal and generalized balance requirements

In the previous subsection we have shown that for $AB$ without any island, the contributions are still given by the ALC proposal. When $AB$ admits an island and no ownerless island appears, then the ALC proposal also applies with the entanglement entropies in the linear combination calculated by the island formula. Nevertheless, in class 3 the situation is different since $\mathrm{Ir}(A)$ is not the island of $A$. It is impossible to write the PEE $\mathcal{I}(A\,\mathrm{Ir}(A), C)$ in terms of the entanglement entropies of subsets. In order to explicitly compute the PEE, we need to find a way to write the PEE in terms of other quantities that are computable.

As we have discussed, in the 2d effective field theory we can compute the two-point correlation function of twist operators, when the two points are settled with reflection symmetry the correlation function computes the entanglement entropy with islands. While when there is no reflection symmetry the physical meaning of the two-point function is not clear due to the self-encoding property of the system. Here we propose that these two-point functions indeed give the PEE between the region enclosed by the two points and the complement of this region. More explicitly, let us consider the twist operators settled at $a$ and $b$ ($b > a$) and denote the connected region $[a, b]$ as $\gamma$. Then, we propose the following equation:

$$\text{Basic proposal 1}: \quad \mathcal{I}(\gamma, \bar{\gamma}) = \tilde{S}_{[a,b]} = \tilde{S}_{(-\infty,a]\cup[b,\infty)}$$
$$= \frac{c}{3}\log\left(\frac{b-a}{\delta}\right) + \frac{c}{6}\varphi(a) + \frac{c}{6}\varphi(b). \tag{41}$$

In non-island phases, the above equation holds as both of the $\mathcal{I}(\gamma, \bar{\gamma})$ and the two-point function give the entanglement entropy of the region $\gamma$. While in island phase, the PEE $\mathcal{I}(\gamma, \bar{\gamma})$ can be classified into the following three classes.

- When $a > 0$ and $\gamma$ does not admit island, the two-point function gives the entanglement of $\gamma$ as in the non-island phase,

$$\mathcal{I}(\gamma, \bar{\gamma}) = S_\gamma\,. \tag{42}$$

- When $a = -b$ and $b > 0$, the two-point function gives the entanglement entropy for the region $A = [0, b]$,

$$\mathcal{I}(\gamma, \bar{\gamma}) = \tilde{S}_\gamma = S_A\,, \tag{43}$$

where $\text{Is}(A) = [-b, 0)$ and $\gamma = A \cup \text{Is}(A)$.

- When $a \neq -b$ and $a < 0$, $\mathcal{I}(\gamma, \bar{\gamma})$ is not the entanglement entropy of any region. By definition it is just the integration (or summation) of two-point PEEs

$$\mathcal{I}(\gamma, \bar{\gamma}) = \int_{x \in \gamma} \int_{y \in \bar{\gamma}} \mathcal{I}(x, y) dx dy\,. \tag{44}$$

Although in the third class $\mathcal{I}(\gamma, \bar{\gamma})$ is not an entanglement entropy, it is also computable following (41). This is crucial since all the PEE $\mathcal{I}(A, B)$ between any two non-overlapping regions, can be written as a linear combination of the special type of PEE between a region and its complement, i.e. $\mathcal{I}(\gamma, \bar{\gamma})$. We will see that, this linear combination of $\mathcal{I}(\gamma, \bar{\gamma})$ has the same structure as the ALC proposal if we also denote $\mathcal{I}(\gamma, \bar{\gamma})$ as

$$\mathcal{I}(\gamma, \bar{\gamma}) \equiv \tilde{S}_\gamma \equiv \tilde{S}_{[a,b]}\,. \tag{45}$$

Later we will also encounter cases with disconnected $\gamma = A \cup \text{Ir}(A)$, where $\text{Ir}(A) = [-d, -c]$, $A = [a, b]$ and $a, b, c, d > 0$. In these cases we propose that

$$\text{Basic proposal 2:} \quad \mathcal{I}(A\text{Ir}(A), \overline{A\text{Ir}(A)}) = \tilde{S}_{[-d,-c] \cup [a,b]} = \tilde{S}_{[-c,a]} + \tilde{S}_{[-d,b]}\,. \tag{46}$$

This is similar to the RT formula for disconnected intervals, which should be a result under the large $c$ limit. However, here we need not compare $\tilde{S}_{[-c,a]} + \tilde{S}_{[-d,b]}$ with $\tilde{S}_{[-d,-c]} + \tilde{S}_{[a,b]}$ and choose the minimal one.

Note that, according to the additivity property, $\mathcal{I}(A\text{Ir}(A), \overline{A\text{Ir}(A)})$ can be written as a linear combination of the type of PEEs $\mathcal{I}(\gamma, \bar{\gamma})$ with connected $\gamma$, which can be computed by the *basic proposal 1*. Let us denote $\overline{A\text{Ir}(A)} = E \cup F$ where the interval $E = [-c, a]$ is sandwiched by $A$ and $\text{Ir}(A)$. Then we have

$$\begin{aligned} \mathcal{I}(A\text{Ir}(A), \overline{A\text{Ir}(A)}) &= \mathcal{I}(AE\,\text{Ir}(A), F) + \mathcal{I}(AF\,\text{Ir}(A), E) - 2\mathcal{I}(E, F) \\ &= \tilde{S}_{[-d,b]} + \tilde{S}_{[-c,a]} - 2\mathcal{I}(E, F)\,, \end{aligned} \tag{47}$$

where we have used the *basic proposal 1* in the second line. The above equation indicates that the two basic proposals are not consistent unless $\mathcal{I}(E, F) = 0$. This is possible since $E$ and $F$ are separated by the region $A$ and its generalized island $\text{Ir}(A)$, which indicates that the entanglement wedge for $E \cup F$ is disconnected. Later we will give a demonstration in support of this statement. *Therefore, the basic proposal 1 given in* (41)*, is the only conjecture we made in this paper to compute PEE.*

Next, we compute a generic PEE based on the basic proposals 1 and 2. For example, let us consider two adjacent intervals $A$ and $B$ and their generalized islands (or reflected islands) $\text{Ir}(A)$ and $\text{Ir}(B)$. Again we denote $C = \overline{A\text{Ir}(A)B\text{Ir}(B)}$. According to the additivity property, the PEE $\mathcal{I}(A\text{Ir}(A), C)$ can be written as

$$\begin{aligned} s_{AB}(A) &= \mathcal{I}(A\text{Ir}(A), C) \\ &= \frac{1}{2}\Big[\mathcal{I}(A\text{Ir}(A)B\text{Ir}(B), C) + \mathcal{I}(A\text{Ir}(A), B\text{Ir}(B)C) - \mathcal{I}(B\text{Ir}(B), A\text{Ir}(A)C)\Big] \\ &= \frac{1}{2}\Big[\tilde{S}_{A\text{Ir}(A)B\text{Ir}(B)} + \tilde{S}_{A\text{Ir}(A)} - \tilde{S}_{B\text{Ir}(B)}\Big]\,. \end{aligned} \tag{48}$$

This is a generalization of the formula (31) with the island Is($A$) and Is($B$) replaced by the reflected islands Ir($A$) and Ir($B$). Interestingly, the above linear combination have the same structure as the ALC proposal (27). The only difference is that, every region appearing in the linear combination has a generalized island companion,

$$S_\gamma \Rightarrow \tilde{S}_{\gamma \, \mathrm{Ir}(\gamma)}. \tag{49}$$

For the non-adjacent case, we need to consider the scenarios where $A$ is sandwiched by two intervals $A_1$ and $A_2$ and compute the contribution

$$s_{A_1 A A_2}(A) = \mathcal{I}(A\,\mathrm{Ir}(A), C), \tag{50}$$

where $C$ is the complement of $A_1 \mathrm{Ir}(A_1) A \mathrm{Ir}(A) A_2 \mathrm{Ir}(A_2)$. According to the additivity property, we can write the contribution in the following way

$$
\begin{aligned}
s_{A_1 A A_2}(A) = \frac{1}{2}\Big[ & \mathcal{I}(A\,\mathrm{Ir}(A)A_1\mathrm{Ir}(A_1), A_2\mathrm{Ir}(A_2)C) + \mathcal{I}(A\,\mathrm{Ir}(A)A_2\mathrm{Ir}(A_2), A_1\mathrm{Ir}(A_1)C) \\
& - \mathcal{I}(A_2\mathrm{Ir}(A_2), \mathcal{I}(A\,\mathrm{Ir}(A)A_1\mathrm{Ir}(A_1)C) - \mathcal{I}(A_1\mathrm{Ir}(A_1), A\,\mathrm{Ir}(A)A_2\mathrm{Ir}(A_2)C)\Big],
\end{aligned} \tag{51}
$$

which can further be written as

$$s_{A_1 A A_2}(A) = \frac{1}{2}\Big[ \tilde{S}_{A\mathrm{Ir}(A)A_1\mathrm{Ir}(A_1)} + \tilde{S}_{A\mathrm{Ir}(A)A_2\mathrm{Ir}(A_2)} - \tilde{S}_{A_2\mathrm{Ir}(A_2)} - \tilde{S}_{A_1\mathrm{Ir}(A_1)} \Big]. \tag{52}$$

It is evident that the above formula coincides with the ALC proposal (8) with the replacement (49).

We call the new formulas (48) and (52) the *generalized ALC proposal* in the island phase. It reduces to (8) when $\mathrm{Ir}(\gamma) = \emptyset$, and reduces to (31) when there is no ownerless islands, i.e. $\mathrm{Ir}(\gamma) = \mathrm{Is}(\gamma)$.

Since the way we compute the contribution is generalized for island phase, the balance requirements should also be modified to a new version in terms of PEEs. For adjacent cases, the generalized balanced requirement is given by

$$\mathcal{I}(A\,\mathrm{Ir}(A), B_1\,\mathrm{Ir}(B_1)) = \mathcal{I}(A_1\,\mathrm{Ir}(A_1), B\,\mathrm{Ir}(B)). \tag{53}$$

For disjoint cases, the two balanced requirements are generalized to be

$$
\begin{aligned}
\mathcal{I}(A_1\mathrm{Ir}(A_1), B\,\mathrm{Ir}(B)B_2\mathrm{Ir}(B_2)) &= \mathcal{I}(B_1\mathrm{Ir}(B_1), A\,\mathrm{Ir}(A)A_2\mathrm{Ir}(A_2)), \\
\mathcal{I}(A\,\mathrm{Ir}(A), B_1\mathrm{Ir}(B_1)B_2\mathrm{Ir}(B_2)) &= \mathcal{I}(B\,\mathrm{Ir}(B), A_1\mathrm{Ir}(A_1)A_2\mathrm{Ir}(A_2)).
\end{aligned} \tag{54}
$$

## 4 Entanglement wedge cross-sections in island phase

### 4.1 Classification for entanglement wedge cross-sections

In this section, we turn to the BPE in island phase and check its correspondence with the EWCS. Before we explicitly solve the balance requirements and compute the BPE, in this section we give a classification for the EWCS of the entanglement wedge of $A \cup B$ in AdS$_3$ bulk with a EoW brane settled at $\rho = \kappa$. The intervals $A$ and $B$ are defined as

$$A: \ [b_1, b_2], \qquad B: \ [b_3, b_4], \qquad 0 \le b_1 < b_2 \le b_3 < b_4. \tag{55}$$

When $b_2 = b_3$ the two intervals are adjacent, while when $b_2 < b_3$ they are disjoint. For both of the adjacent and disjoint cases, we classify the EWCS in the following.

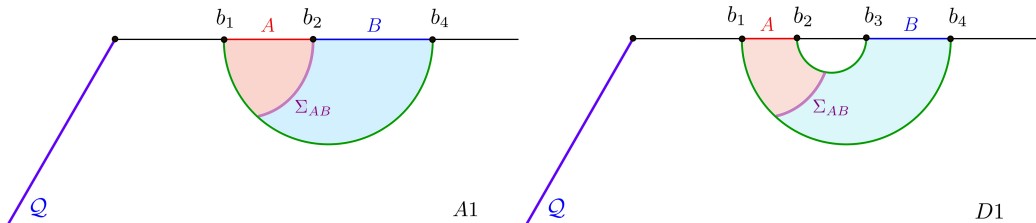

Figure 3: Entanglement wedge cross-sections for $A \cup B$ admitting no island.

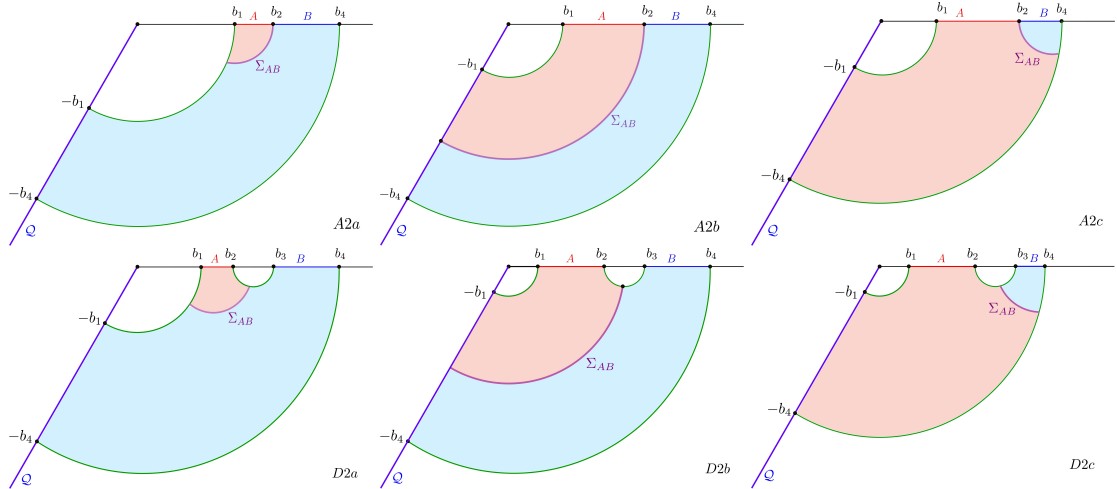

Figure 4: Entanglement wedge cross-sections for $A \cup B$ admitting an island.

**Phase A1 and D1**: Configurations when $A \cup B$ or $[b_1, b_4]$ does not admit an island.

In this phase, the entanglement wedge of $AB$ is the same as the one in non-island phase, as well as the EWCS $\Sigma_{AB}$ (see Fig.3). Also the area of the EWCS has been studied in [50] and are given by,

$$\textbf{Phase-A1}: \frac{\text{Area}[\Sigma_{AB}]}{4G_N} = \frac{c}{6} \log \frac{2(b_2 - b_1)(b_4 - b_2)}{\delta(b_4 - b_1)}, \tag{56}$$

$$\textbf{Phase-D1}: \frac{\text{Area}[\Sigma_{AB}]}{4G_N} = \frac{c}{6} \log \frac{1 + \sqrt{x}}{1 - \sqrt{x}}, \quad x = \frac{(b_4 - b_3)(b_2 - b_1)}{(b_3 - b_1)(b_4 - b_2)}. \tag{57}$$

**Phase A2 and D2**: Configurations where $AB$ or $[b_1, b_4]$ admits an island.

In these configurations the RT surface $\mathcal{E}_{AB}$ homologous to $[b_1, b_4]$ becomes disconnected[10] (see Fig.4), and is decomposed in two pieces

$$\mathcal{E}_{AB} = RT(b_1) \cup RT(b_4), \tag{58}$$

where, for example, $RT(b_1)$ denotes the piece that emanates from $x = b_1$ and lands at the bulk (cutoff) brane. Note that there are three saddle points for the area of the entanglement wedge cross-section, as one of the endpoints of $\Sigma_{AB}$ can anchor on three choices: $RT(b_1)$, the brane, and $RT(b_2)$. One should choose the EWCS with the minimal area. Nevertheless, any of the three choices can be the minimal one if we adjust the four parameters $b_i$ properly. So these configurations can further be classified into three different phases for which the areas of the EWCS have been studied in [50, 106, 109] and these are given by:

---

[10]Note that for disjoint $A$ and $B$, the other part of the RT surface $\mathcal{E}_{AB}$ homologous to $[b_2, b_3]$ remains connected, as depicted in Fig.4.

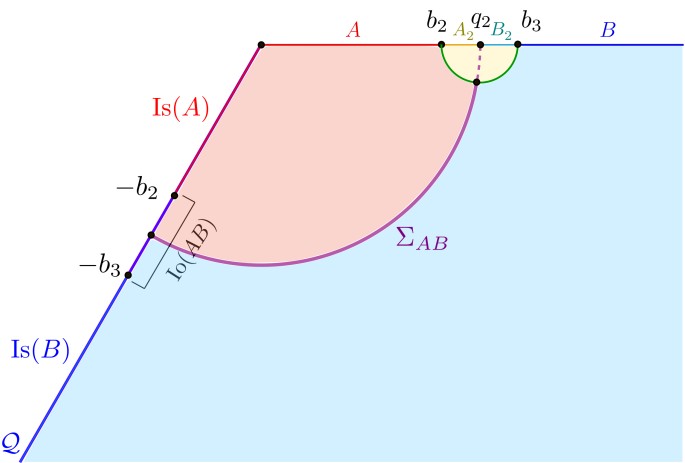

Figure 5: A typical configuration: $A = [0, b_2]$, $B = [b_3, +\infty]$ and their sandwiched interval $[b_2, b_3]$ admits no island. There is an ownerless island $\mathrm{Io}(AB) = [-b_3, -b_2]$.

1. $\Sigma_{AB}$ is anchored on $\mathrm{RT}(b_1)$, the RT surface connecting $b_1$ and the brane:

$$\textbf{Phase A2a}: \frac{\mathrm{Area}[\Sigma_{AB}]}{4G_N} = \frac{c}{6} \log \frac{b_2^2 - b_1^2}{b_1 \delta}, \tag{59}$$

$$\textbf{Phase D2a}: \frac{\mathrm{Area}[\Sigma_{AB}]}{4G_N} = \frac{c}{6} \log \frac{b_2 b_3 - b_1^2 + \sqrt{(b_2^2 - b_1^2)(b_3^2 - b_1^2)}}{b_1(b_3 - b_2)}. \tag{60}$$

2. $\Sigma_{AB}$ is anchored on the brane:

$$\textbf{Phase A2b}: \frac{\mathrm{Area}[\Sigma_{AB}]}{4G_N} = \frac{c}{6} \log \frac{2b_2}{\delta} + \frac{c}{6}\kappa, \tag{61}$$

$$\textbf{Phase D2b}: \frac{\mathrm{Area}[\Sigma_{AB}]}{4G_N} = \frac{c}{6} \log \frac{\sqrt{b_3} + \sqrt{b_2}}{\sqrt{b_3} - \sqrt{b_2}} + \frac{c}{6}\kappa. \tag{62}$$

3. $\Sigma_{AB}$ is anchored on $\mathrm{RT}(b_4)$:

$$\textbf{Phase A2c}: \frac{\mathrm{Area}[\Sigma_{AB}]}{4G_N} = \frac{c}{6} \log \frac{b_4^2 - b_2^2}{b_4 \delta}, \tag{63}$$

$$\textbf{Phase D2c}: \frac{\mathrm{Area}[\Sigma_{AB}]}{4G_N} = \frac{c}{6} \log \frac{b_2 b_3 - b_4^2 + \sqrt{(b_2^2 - b_4^2)(b_3^2 - b_4^2)}}{b_4(b_3 - b_2)}. \tag{64}$$

4. Configurations where the entanglement wedge of $AB$ becomes disconnected, hence EWCS disappears, i.e.

$$\textbf{Phase D3}: \frac{\mathrm{Area}[\Sigma_{AB}]}{4G_N} = 0. \tag{65}$$

## 4.2 A naive calculation of BPE with ownerless islands

Now we show that for the configurations with ownerless islands, if we insist on applying the ALC proposal to calculate the PEE, then the resulting BPE does not match with the EWCS. To be specific, let us consider a typical configuration (see Fig.5) where $A = [0, b_2]$, $B = [b_3, +\infty]$

and their sandwiched interval admits no island. In this case Is($AB$) covers the whole $x < 0$ region, and there appears an ownerless island

$$\text{Io}(AB) = [-b_3, -b_2].$$ (66)

The partition point $x = q_2$ divides the sandwiched interval into $A_2 = [b_1, q_2]$ and $B_2 = [q_2, b_3]$. As discussed earlier, the appearance of the ownerless island makes the contributions coming from the individual entanglement entropies complicated, and the naive application of the ALC proposal does not suffice. If we naively apply the ALC proposal to calculate the PEE, then we have

$$s_{AA_2}(A) = \frac{1}{2}(S_{AA_2} + S_A - S_{A_2}) = \frac{c}{12} \log \frac{4q_2 b_2}{(q_2 - b_2)^2} + \frac{c}{6} \kappa,$$ (67)

$$s_{BB_2}(B) = \frac{1}{2}(S_{BB_2} + S_B - S_{B_2}) = \frac{c}{12} \log \frac{4q_2 b_3}{(b_3 - q_2)^2} + \frac{c}{6} \kappa.$$ (68)

Solving the balance condition $s_{AA_2}(A) = s_{BB_2}(B)$ we find

$$q_2 = \sqrt{b_2 b_3}.$$ (69)

Plugging the above $q_2$ into the PEE, we find that the BPE is given by

$$\begin{aligned} \text{BPE} = s_{AA_2}(A)|_{\text{balanced}} &= \frac{c}{6} \log \frac{2\sqrt{\sqrt{b_2 b_3}}}{\sqrt{b_3} - \sqrt{b_2}} + \frac{c}{6} \kappa \\ &< \frac{c}{6} \log \frac{\sqrt{b_3} + \sqrt{b_2}}{\sqrt{b_3} - \sqrt{b_2}} + \frac{c}{6} \kappa = \frac{\text{Area}[\Sigma_{AB}]}{4G_N}. \end{aligned}$$ (70)

Clearly, the BPE calculated in this way does not match with the area of the EWCS.

Let us take a deeper look at the $s_{AA_2}(A)$ constructed from the ALC proposal and write it in terms of the two-body-correlation representation. We find

$$\begin{aligned} s_{AA_2}(A) &= \frac{1}{2}(S_{AA_2} + S_A - S_{A_2}) \\ &= \frac{1}{2}\Big[ \mathcal{I}(AA_2 \text{Is}(AA_2), BB_2 \text{Is}(BB_2)) + \mathcal{I}(A\text{Is}(A), BB_2 A_2 \text{Is}(BB_2 A_2)) - \mathcal{I}(A_2, AB \text{Is}(AB) \cup B_2) \Big] \\ &= \mathcal{I}(A\text{Is}(A), BB_2 \text{Is}(BB_2)) + \frac{1}{2}\Big[ \mathcal{I}(\text{Io}(A_2), BB_2 \text{Is}(BB_2) \cup A) - \mathcal{I}(A_2, \text{Io}(A)) \Big], \end{aligned}$$ (71)

where we used the additivity of the PEE and Is($AB$) = Is($A$)Is($B$)Io($A$)Io($B$). This result does not look like a contribution from $A$ to the entropy $S_{AA_2}$ in any sense. The contribution from the ownerless island is not properly taken into account.

Here we give a glimpse of the correct way to calculate $s_{AA_2}(A)$ when the ownerless island regions appear. At first we decompose the islands Is($AA_2$) and Is($BB_2$) in the following way

$$\text{Is}(AA_2) = \text{Is}(A) \cup \text{Io}(A), \qquad \text{Is}(BB_2) = \text{Is}(B) \cup \text{Io}(B),$$ (72)

where the ownerless island regions are assigned to $s_{AA_2}(A)$ and $s_{BB_2}(B)$ respectively and are chosen to be

$$\text{Io}(B) = [-b_3, -q_2], \qquad \text{Io}(A) = [-q_2, -b_2].$$ (73)

The above ownerless island regions Io($A$) and Io($B$) are determined by the balance requirement that gives the minimal BPE. This will be explained in the later sections. Then using the

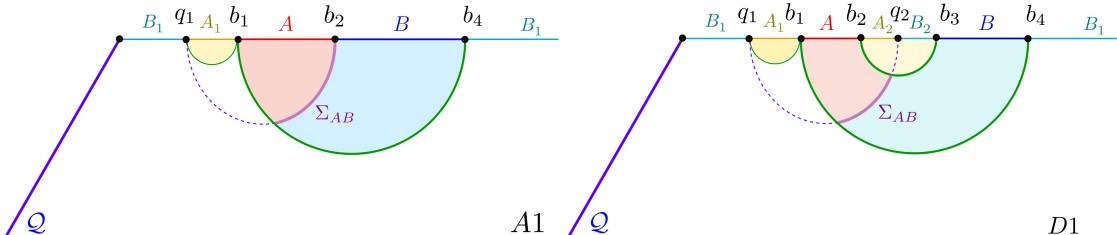

Figure 6: The geometric picture for the BPE in Phase A1 and D1. In this case, the partition point $x = q_1$ is settled either in the region $x < b_1$ (including the island region $x < 0$) or the region $x > b_4$.

generalized ALC formula, we have

$$
\begin{aligned}
s_{AA_2}(A) &= \frac{1}{2}\Big[\tilde{S}_{A\mathrm{Is}(A)\mathrm{Io}(A)A_2} + \tilde{S}_{A\mathrm{Is}(A)\mathrm{Io}(A)} - \tilde{S}_{A_2}\Big] \\
&= \frac{c}{6}\log\frac{\sqrt{b_3}+\sqrt{b_2}}{\sqrt{b_3}-\sqrt{b_2}} + \frac{c}{6}\kappa\,,
\end{aligned}
\tag{74}
$$

which exactly coincide with the EWCS. One can check that the balance requirement $s_{AA_2}(A) = s_{BB_2}(B)$ is satisfied by choosing (73).

# 5 Balanced partial entanglement in island phases

In this section we consider the BPE in island phase and their correspondence with the EWCS. Compared with the non-island phase, the phase space for the EWCS in island phase is more complicated as there are more saddles. Also there are new phase transitions for the EWCS. These indicate that the phase space of the BPE should also be more complicated, and checking its correspondence with the EWCS becomes more challenging in island phase.

For each configuration, there are different ways to assign the contribution from the ownerless island region. We consider all the possible assignments of the ownerless island region and subsequently solve the balance requirements for each assignment. For all the solutions we compute the corresponding BPEs, and choose the minimal one if there are multiple solutions to the balance requirements. Remarkably, we find that the BPEs calculated under different assignments of the ownerless island regions exactly correspond to the different saddles of the EWCS. Also the minimal BPE identically matches the minimal EWCS and hence the correspondence between the BPE and the EWCS holds in island phase. In the following we conduct the analysis for the BPE in all the configurations we have classified in the previous section.

## 5.1 *AB* with no island

Let us first consider phases-A1 where $A$ and $B$ are adjacent and $A\cup B$ admits no island. In this case none of $A$ and $B$ or $A\cup B$ admit islands or their generalized counterparts and therefore no degrees of freedom outside $AB$ contribute to the mixed state correlation between $A$ and $B$. Let us divide the complement of $AB$ into $A_1\cup B_1$ with the partition point $x = q_1$ (see the left panel in Fig.6). Note that, in this case the island region $x < 0$ is included in $A_1B_1$. Based on this

partition in phase-A1, the balance requirement is the same as (11) in no-island phase, where

$$\mathcal{I}(A, BB_1) = \frac{1}{2}[\mathcal{I}(AA_1, BB_1) + \mathcal{I}(A, BB_1 A_1) - \mathcal{I}(A_1, BB_1 A)] = \frac{c}{6} \log \frac{(b_2 - q_1)(b_2 - b_1)}{\delta(b_1 - q_1)}, \quad (75)$$

$$\mathcal{I}(B, AA_1) = \frac{1}{2}[\mathcal{I}(AA_1, BB_1) + \mathcal{I}(B, AA_1 B_1) - \mathcal{I}(B_1, AA_1 B)] = \frac{c}{6} \log \frac{(b_2 - q_1)(b_4 - b_2)}{\delta(b_4 - q_1)}. \quad (76)$$

Solving the balance condition $\mathcal{I}(A, BB_1) = \mathcal{I}(B, AA_1)$ we find a unique solution

$$q_1 = \frac{2b_1 b_4 - b_1 b_2 - b_2 b_4}{b_1 + b_4 - 2b_2}. \quad (77)$$

The corresponding BPE is given by

$$\text{BPE}(A : B) = \mathcal{I}(A, BB_1)|_{balanced} = \frac{c}{6} \log \frac{2(b_2 - b_1)(b_4 - b_2)}{\delta(b_4 - b_1)}, \quad (78)$$

which is exactly the area of EWCS given in (56).

For phase-D1, there are two partition points $q_1$ and $q_2$ which divide the complement of $AB$ into four regions $A_1 B_1 \cup A_2 B_2$ (see the right panel in Fig.6). Also in this case $AB$ admits no island. Using the two-body-correlation representation we have

$$\mathcal{I}(A, BB_1 B_2) = \frac{1}{2}[\mathcal{I}(A_1 A, A_2 BB_1 B_2) - \mathcal{I}(A_1, AA_2 BB_1 B_2) + \mathcal{I}(A_2 A, A_1 BB_1 B_2) - \mathcal{I}(A_2, AA_1 BB_1 B_2)]$$
$$= \frac{c}{6} \log \frac{(b_2 - q_1)(q_2 - b_1)}{(b_1 - q_1)(q_2 - b_2)}, \quad (79)$$

$$\mathcal{I}(B, AA_1 A_2) = \frac{1}{2}[\mathcal{I}(B_1 B, B_2 AA_2 A_1) - \mathcal{I}(B_1, BB_2 AA_2 A_1) + \mathcal{I}(B_2 B, B_1 AA_2 A_1) - \mathcal{I}(B_2, BB_1 AA_2 A_1)]$$
$$= \frac{c}{6} \log \frac{(b_3 - q_1)(b_4 - q_2)}{(b_4 - q_1)(b_3 - q_2)}. \quad (80)$$

and

$$\mathcal{I}(A_2, BB_1 B_2) = \frac{1}{2}[\mathcal{I}(A_1 AA_2, BB_1 B_2) - \mathcal{I}(A_1 A, A_2 BB_1 B_2) + \mathcal{I}(A_2, A_1 ABB_1 B_2)]$$
$$= \frac{c}{6} \log \frac{(q_2 - q_1)(q_2 - b_2)}{\delta(b_2 - q_1)}, \quad (81)$$

$$\mathcal{I}(B_2, AA_1 A_2) = \frac{1}{2}[\mathcal{I}(B_1 BB_2, AA_2 A_1) - \mathcal{I}(B_1 B, B_2 AA_2 A_1) + \mathcal{I}(B_2, B_1 BAA_2 A_1)]$$
$$= \frac{c}{6} \log \frac{(q_2 - q_1)(b_3 - q_2)}{\delta(b_3 - q_1)}. \quad (82)$$

Solving the two balance conditions $\mathcal{I}(A, BB_1 B_2) = \mathcal{I}(A, BB_1 B_2)$ and $\mathcal{I}(A_2, BB_1 B_2) = \mathcal{I}(B_2, AA_1 A_2)$, we determine the two partition points as follows

$$q_1 = \frac{b_1 b_4 - b_2 b_3 - \sqrt{Y}}{b_4 - b_3 - b_2 + b_1}, \qquad q_2 = \frac{b_1 b_4 - b_2 b_3 + \sqrt{Y}}{b_4 - b_3 - b_2 + b_1},$$
$$Y \equiv (b_1 - b_2)(b_1 - b_3)(b_2 - b_4)(b_3 - b_4). \quad (83)$$

Then the BPE may be computed by substituting the above balanced partition points into the corresponding PEE as follows

$$\text{BPE}(A : B) = \mathcal{I}(A, BB_1 B_2)|_{balanced} = \frac{c}{6} \log \frac{1 + \sqrt{x}}{1 - \sqrt{x}}, \quad x = \frac{(b_4 - b_3)(b_2 - b_1)}{(b_3 - b_1)(b_4 - b_2)}, \quad (84)$$

which is also identical to the area of EWCS (57).

Before we go ahead, we would like to comment on the BPE calculated via the contribution representation in this case. For example, in Phase A1 we can calculate $s_{AA_1}(A)$ and $s_{BB_1}(B)$ and then apply the balance requirement $s_{AA_1}(A) = s_{BB_1}(B)$ to determine $q_1$. One can start from some $b_2$ close to $b_1$ such that the $AA_1$ also admits no island. In this case the island $Is(BB_1)$ covers the whole $x < 0$ region and there is a ownerless island region $Io(BB_1) = [-b_4, q]$ for $BB_1$. We can find a solution for the balance requirement when we assign the ownerless island region to $B_1$ such that $Ir(B_1) = (-\infty, 0)$ and $Ir(B) = \emptyset$. Remarkably, the balance point and BPE calculated in this way coincide with our previous results.

However, when $b_2$ goes farther from $b_1$, the balance solution without any island for $AA_1$ no longer exists, hence we should consider the $AA_1$ admitting island. As a result we have $Ir(A_1) \neq \emptyset$. In this case no matter how we assign the contributions from the ownerless island regions $Io(AA_1)$ and $Io(BB_1)$, the solutions to the balance requirements do not exist. So our previous results are the unique solution to the balance requirements. This is also consistent with the observation that, the EWCS is not a portion of the RT surface for any $AA_1$ which admits island. Therefore, we may conclude that the two-body-correlation representation of the balance requirements includes partition configurations that cannot be described by the contribution representation.

## 5.2 Adjacent $AB$ with island

In this subsection, we consider the phase-A2 where $A = [b_1, b_2]$ and $B = [b_2, b_4]$ and $A \cup B$ has an entanglement island. Here the regions $A$ or $B$ individually may or may not admit their own islands. We consider the following three configurations with given assignments for the ownerless island region $Io(AB)$:

1. A2a: $Ir(A) = \emptyset$, $Ir(B) = [-b_4, -b_1]$,

2. A2b: $Ir(A) = [-q, -b_1]$, $Ir(B) = [-b_4, -q]$,

3. A2c: $Ir(A) = [-b_4, -b_1]$, $Ir(B) = \emptyset$.

Note that, there are configurations with $Ir(A) \subset Is(A)$ (or $Ir(B) \subset Is(B)$) when $Is(A) \neq \emptyset$ (or $Is(B) \neq \emptyset$) for certain choice of $q$. If we take these configurations seriously and solve the balance requirements to get a BPE, then we get the result for the no-island phase. Also this BPE will not be the minimal one. For simplicity, we only consider the cases with $Is(A) \subset Ir(A), Is(B) \subset Ir(B)$ by properly choosing the partition point $x = -q$. Nevertheless, as we will see, these extra configurations can not give the minimal BPE. In the following, we will systematically solve the balance requirements for each assignment and obtain the corresponding BPE.

### 5.2.1 Phase-A2a

For $Ir(A) = \emptyset$ and $Ir(B) = [-b_4, -b_1]$, there is no island contribution to $A$ and all of the island $Is(AB)$ contributes to $B$. Let us assume that the partition point $q_1$ lies at $0 < q_1 < b_1$ such that $A_1 = [q_1, b_1]$, $B_1 = [0, q_1] \cup [b_4, \infty)$. Furthermore we assume $AA_1$ does not admit an island and hence $A_1$ also does not receive any island contribution, i.e.

$$Ir(A_1) = \emptyset, \quad Ir(B_1) = [-b_1, 0] \cup (-\infty, -b_4]. \tag{85}$$

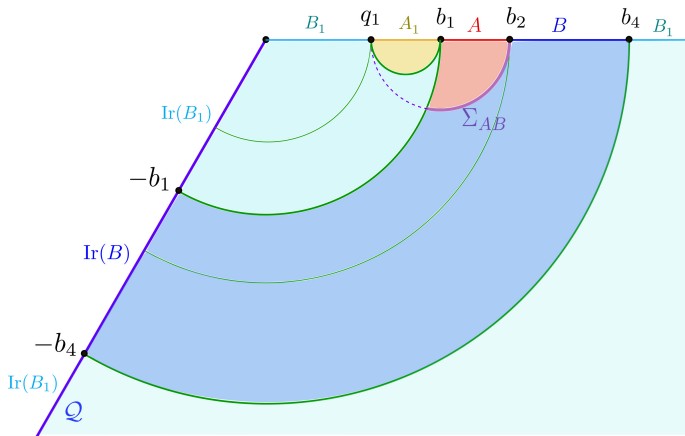

Figure 7: Phase-A2a: $\text{Ir}(A) = \emptyset, \text{Ir}(B) = \text{Is}(AB) = [-b_4, -b_1]$.

The schematics of the setup is depicted in Fig.7. In this configuration, we compute the following two PEEs via the generalized ALC proposal

$$
\begin{aligned}
\mathcal{I}(A, B\text{Ir}(B) \cup B_1\text{Ir}(B_1)) &= \frac{1}{2}\Big[\tilde{S}_{AA_1} + \tilde{S}_A - \tilde{S}_{A_1}\Big] \\
&= \frac{1}{2}(\tilde{S}_{[q_1,b_2]} + \tilde{S}_{[b_1,b_2]} - \tilde{S}_{[q_1,b_1]}) \\
&= \frac{c}{6} \log \frac{(b_2 - q_1)(b_2 - b_1)}{\delta(b_1 - q_1)},
\end{aligned}
\tag{86}
$$

and

$$
\begin{aligned}
\mathcal{I}(B\text{Ir}(B), AA_1) &= \frac{1}{2}\Big[\tilde{S}_{B\text{Ir}(B)B_1\text{Ir}(B_1)} + \tilde{S}_{B\text{Ir}(B)} - \tilde{S}_{B_1\text{Ir}(B_1)}\Big] \\
&= \frac{1}{2}(\tilde{S}_{[q_1,b_2]} + \tilde{S}_{[-b_4,-b_1]\cup[b_2,b_4]} - \tilde{S}_{[-\infty,-b_4]\cup[-b_1,q_1]\cup[b_4,\infty]}) \\
&= \frac{c}{6} \log \frac{(b_2 - q_1)(b_2 + b_1)}{(q_1 + b_1)\delta}.
\end{aligned}
\tag{87}
$$

In (87) we have used the *basic proposal 2* to obtain,

$$
\tilde{S}_{[-b_4,-b_1]\cup[b_2,b_4]} = \tilde{S}_{[-b_1,b_2]} + \tilde{S}_{[-b_4,b_4]},
\tag{88}
$$

$$
\tilde{S}_{[-\infty,-b_4]\cup[-b_1,q_1]\cup[b_4,\infty]} = \tilde{S}_{[-b_4,-b_1]\cup[q_1,b_4]} = \tilde{S}_{[-b_1,q_1]} + \tilde{S}_{[-b_4,b_4]}.
\tag{89}
$$

Solving the balance condition $\mathcal{I}(A, B\text{Ir}(B) \cup B_1\text{Ir}(B_1)) = \mathcal{I}(B\text{Ir}(B), AA_1)$, we find the condition

$$
q_1 = \frac{b_1^2}{b_2}.
\tag{90}
$$

Plugging $q_1 = b_1^2/b_2$ back in (86) or (87), we immediately find the BPE as follows

$$
\text{BPE}(A, B) = \mathcal{I}(A, B\text{Ir}(B) \cup B_1\text{Ir}(B_1))|_{balanced} = \frac{c}{6} \log \frac{b_2^2 - b_1^2}{b_1 \delta}.
\tag{91}
$$

This result coincide with the area of the EWCS saddle in (59).

One can then consider the possibility that $AA_1$ admits an island. In this case $\text{Ir}(A_1) \neq \emptyset$ and we should consider other configurations of $\text{Ir}(A_1)$ and $\text{Ir}(B_1)$. Nevertheless, the solution to the balance requirement does not exist in such a configuration. So the result (91) is the only solution for the configurations A2a.

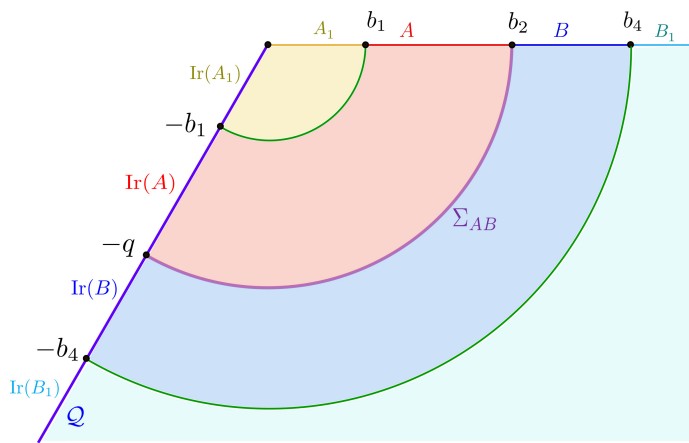

Figure 8: Phase-A2b: $\text{Ir}(A) = [-q, -b_1], \text{Ir}(B) = [-b_4, -q]$.

### 5.2.2 Phase-A2b

In this case, as depicted in Fig.8, we have the generalized islands $\text{Ir}(A) = [-q, -b_1]$ and $\text{Ir}(B) = [-b_4, -q]$, where $x = -q$ is the partition point of $\text{Is}(AB) = \text{Ir}(A) \cup \text{Ir}(B)$. Also we can choose

$$A_1 = [0, b_1], \quad B_1 = [b_4, \infty]. \tag{92}$$

In this configuration, we have $\text{Is}(A_1) \cup \text{Is}(B_1) = \text{Is}(A_1 B_1)$, hence there are no ownerless island regions for $A_1 B_1$ and

$$\text{Ir}(A_1) = \text{Is}(A_1) = [-b_1, 0), \qquad \text{Ir}(B_1) = \text{Is}(B_1) = [-\infty, -b_4]. \tag{93}$$

Then we calculate the following PEEs via the generalized ALC proposal

$$
\begin{aligned}
\mathcal{I}(A\text{Ir}(A), B\text{Ir}(B)B_1\text{Ir}(B_1)) &= \frac{1}{2}\Big[\tilde{S}_{A\text{Ir}(A)A_1\text{Ir}(A_1)} + \tilde{S}_{A\text{Ir}(A)} - \tilde{S}_{A_1\text{Ir}(A_1)}\Big] \\
&= \frac{1}{2}\Big[\tilde{S}_{[-q,b_2]} + \tilde{S}_{[-q,-b_1]\cup[b_1,b_2]} - \tilde{S}_{[-b_1,b_1]}\Big] \\
&= \tilde{S}_{[-q,b_2]},
\end{aligned}
\tag{94}
$$

$$
\begin{aligned}
\mathcal{I}(B\text{Ir}(B), A\text{Ir}(A)A_1\text{Ir}(A_1)) &= \frac{1}{2}\Big[\tilde{S}_{B\text{Ir}(B)B_1\text{Ir}(B_1)} + \tilde{S}_{B\text{Ir}(B)} - \tilde{S}_{B_1\text{Ir}(B_1)}\Big] \\
&= \frac{1}{2}\Big[\tilde{S}_{[-q,b_2]} + \tilde{S}_{[-b_4,-q]\cup[b_2,b_4]} - \tilde{S}_{[-b_4,b_4]}\Big] \\
&= \tilde{S}_{[-q,b_2]}.
\end{aligned}
$$

It is interesting that the balance requirement

$$\mathcal{I}(A\text{Ir}(A), B\text{Ir}(B)B_1\text{Ir}(B_1)) = \mathcal{I}(B\text{Ir}(B), A\text{Ir}(A)A_1\text{Ir}(A_1)), \tag{95}$$

is satisfied for all the choice of $q$ if $b_1 < q < b_4$. Since different choices of $q$ give us different BPE, we should choose the minimal one according to the minimal requirement. More explicitly we choose the $q$ that satisfy

$$\partial_q \tilde{S}_{[-q,b_2]} = \partial_q \left[\frac{c}{3}\log\left(\frac{b_2 + q}{\delta}\right) + \frac{c}{6}(\kappa - \log\frac{2q}{\delta})\right] = 0, \tag{96}$$

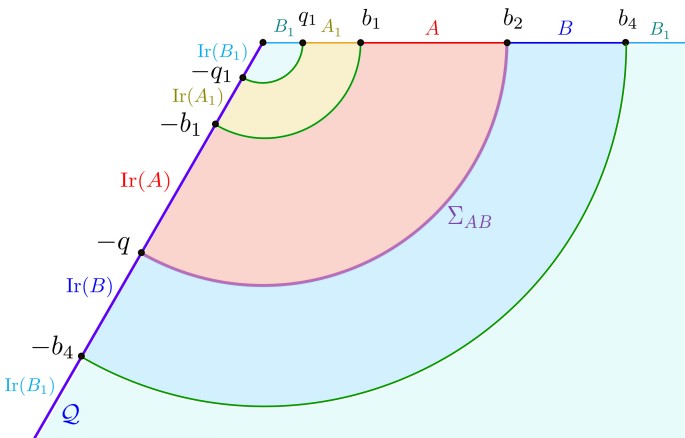

Figure 9: Another possible configuration for the phase-A2b: $\text{Ir}(A_1) = [-b_1, -q_1]$, $\text{Ir}(B_1) = [-q_1, 0] \cup (-\infty, -b_4]$.

which has a simple solution,

$$q = b_2. \tag{97}$$

The choice $q = b_2$ gives the expected minimal BPE

$$\text{BPE}(A, B) = \tilde{S}_{[-q, b_2]}|_{minimal} = \frac{c}{6} \log \frac{2b_2}{\delta} + \frac{c}{6} \kappa, \tag{98}$$

which coincide with the EWCS saddle (61) that is anchored on the EoW brane.

### 5.2.3 The vanishing PEE in island phase

One can also consider other configurations for the phase A2b, for example where a portion of $A_1$ is transferred to $B_1$ compared with the previous configuration (see Fig.9),

$$A_1 = [q_1, b_1], \qquad B_1 = [0, q_1] \cup [b_4, \infty]. \tag{99}$$

If $A_1$ admit island, i.e. $q_1 \leq r^* b_1$, then we have

$$\text{Ir}(A_1) = \text{Is}(A_1) = [-b_1, -q_1], \qquad \text{Ir}(B_1) = \text{Is}(B_1) = [-q_1, 0] \cup [-\infty, -b_4]. \tag{100}$$

In these configurations, we may also set $q = b_2$ such that $\text{Ir}(A)$ and $\text{Ir}(B)$ are not changed. Subsequently, we find that the PEEs are given by

$$
\begin{aligned}
\mathcal{I}(A\text{Ir}(A), B\text{Ir}(B)B_1\text{Ir}(B_1)) &= \frac{1}{2}\Big[ \tilde{S}_{A\text{Ir}(A)A_1\text{Ir}(A_1)} + \tilde{S}_{A\text{Ir}(A)} - \tilde{S}_{A_1\text{Ir}(A_1)} \Big] \\
&= \frac{1}{2}\Big[ \tilde{S}_{[-b_2,q_1]\cup[q_1,b_2]} + \tilde{S}_{[-b_2,-b_1]\cup[b_1,b_2]} - \tilde{S}_{[-b_1,-q_1]\cup[q_1,b_1]} \Big] \\
&= \tilde{S}_{[-b_2,b_2]}, \\
\mathcal{I}(B\text{Ir}(B), A\text{Ir}(A)A_1\text{Ir}(A_1)) &= \frac{1}{2}\Big[ \tilde{S}_{B\text{Ir}(B)B_1\text{Ir}(B_1)} + \tilde{S}_{B\text{Ir}(B)} - \tilde{S}_{B_1\text{Ir}(B_1)} \Big] \\
&= \frac{1}{2}\Big[ \tilde{S}_{[q_1,b_3]\cup[-b_3,-q_1]} + \tilde{S}_{[-b_4,-b_2]\cup[b_2,b_4]} - \tilde{S}_{[q_1,b_4]\cup[-b_4,-q_1]} \Big] \\
&= \tilde{S}_{[-b_2,b_2]}.
\end{aligned}
\tag{101}
$$

It is obvious that, such kind of configurations with $q = b_2$ and $A_1$ admitting an island satisfy the balance requirements and give the same BPE as in (98). This means that the balance point

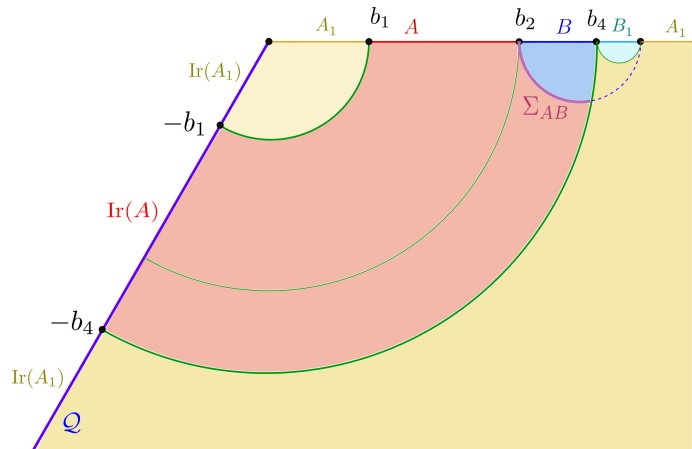

Figure 10: Phase-A2c: $\text{Ir}(A) = \text{Is}(AB) = [-b_4, -b_1], \text{Ir}(B) = \emptyset$.

that gives the minimal BPE is highly non-unique. In other words, it does not change the BPE whether we assign the region $E = [-q_1, q_1]$ to $A_1\text{Ir}(A_1)$ or $B_1\text{Ir}(B_1)$, which indicates that the PEE between the regions $E$ and $A\text{Ir}(A)B\text{Ir}(B)$ is zero,

$$\mathcal{I}(E, AB\text{Ir}(AB)) = 0. \tag{102}$$

Note that, when $q = b_2$ the configurations has reflection symmetry and there is no ownerless islands. Also the calculation only involves entanglement entropies for regions with islands, hence the results do not rely on the two basic proposals (41) and (46). This is important as it indicates that the vanishing PEE (102) can be derived independently from the basic proposals for PEE.

Furthermore, we can take the limit $b_4 \to \infty$ and denote $F = AB\text{Ir}(AB) = [-\infty, -b_1] \cup [b_1, \infty]$. Under this limit, the configuration is always in Phase-A2b with a proper choice for $b_2$, and the equation (102) still holds. So we can make the following generic statement:

- For any region $E$ and $F$ which is separated by a region $A$ and its (generalized) island $\text{Ir}(A)$, we have

$$\mathcal{I}(E, F) = 0, \tag{103}$$

  which is crucial for us to derive (46) from (41). According to the additivity and positivity properties, the PEE between any subsets of $E$ and $F$ also vanishes.

### 5.2.4 Phase-A2c

For the assignment $\text{Ir}(A) = [-b_4, -b_1]$ and $\text{Ir}(B) = \emptyset$, the analysis is symmetric to the case of Phase-A2a in the exchange of $A$ and $B$. In this case, assuming that $BB_1$ does not admit an island, we should solve the balance requirement $\mathcal{I}(A\text{Ir}(A), BB_1) = \mathcal{I}(B, A\text{Ir}(A)A_1\text{Ir}(A_1))$. We can find a solution satisfying both the balance requirement and the minimal requirement, which gives the partition point $x = q_1 > b_4$ as follows,

$$q_1 = \frac{b_4^2}{b_2}. \tag{104}$$

The BPE in this case is given by

$$\text{BPE}(A : B) = \frac{c}{6} \log \frac{b_4^2 - b_2^2}{b_4 \delta}, \tag{105}$$

which exactly matches with the EWCS saddle (63) that is anchored on the piece of the RT surface $\mathcal{E}_{AB}$ emanating from $x = b_4$.

### 5.2.5 Minimizing the BPE in Phase-A2

Now we compare the three BPEs in the phase-A2, which correspond to the three saddle EWCSs, and choose the minimal one. When we computed the BPE in Phase-A2a, we assumed that $AA_1$ does not admit an island and hence have not considered the possibility that there may be balance points when $AA_1$ admits island. Here we can exclude this possibility by showing that when the Phase-A2a gives smaller BPE than the Phase-A2b, $AA_1$ should not admit island. A similar statement also applies to $BB_1$ in Phase-A2c.

In the case of phase-A2a, we know $A$ does not admit an island and hence $b_1/b_2 > r^*$. Since the solution $q_1 = b_1^2/b_2$ can be written as

$$\frac{q_1}{b_1} = \frac{b_1}{b_2} > r^*, \tag{106}$$

we conclude that $A_1$ also does not admit an island. Nevertheless the assumption that $AA_1$ does not admit island may not be satisfied. Now we compare the BPE in Phase A2a and A2b, and find that the critical point between these two phases is

$$b_1 = (\sqrt{1 + e^{2\kappa}} - e^{\kappa})b_2 = \sqrt{r^*}b_2. \tag{107}$$

When $b_2 < b_1/\sqrt{r^*}$ the Phase-A2a gives smaller BPE, and furthermore we have

$$q_1 = b_1^2/b_2 > r^*b_2, \tag{108}$$

which confirms our assumption that $AA_1$ does not admit island.

Similarly one can compare the BPE in Phase-A2b and Phase-A2c and find the critical point to be

$$b_2 = \sqrt{r^*}b_4. \tag{109}$$

When $b_4 < b_2/\sqrt{r^*}$ the Phase-A2c gives smaller BPE, and furthermore we have

$$q_1 = b_4^2/b_2 < r^*b_4, \tag{110}$$

which confirms our assumption that $BB_1$ does not admit island.

When $AB$ admits an island, we require $b_1 \leq r^*b_4$ and we can always find a $b_2$ inside $(b_1, b_4)$ satisfying the following inequality

$$b_1/\sqrt{r^*} \leq b_2 \leq \sqrt{r^*}b_4. \tag{111}$$

In other words, we can confirm that when $AB$ admits island, there always exists a $b_2$ such that the BPE or EWCS is in Phase-A2b.

## 5.3 Disjoint $AB$ with island

When $A = [b_1, b_2]$ and $B = [b_3, b_4]$ are disjoint, the EWCS $\Sigma_{AB} \neq 0$ only when their sandwiched interval $[b_2, b_3]$ has no island. Let us denote the partition point inside this sandwiched interval as $x = q_2$ which divides $[b_2, b_3]$ into $A_2 \cup B_2$. Similar to the adjacent phases, according to whether $\text{Ir}(A)$ and $\text{Ir}(B)$ exist, we have three sub-phases:

1. D2a: $\text{Ir}(A) = \emptyset, \text{Ir}(B) = [-b_4, -b_1]$,

2. D2b: $\text{Ir}(A) = [-q, -b_1], \text{Ir}(B) = [-b_4, -q]$,

3. D2c: $\text{Ir}(A) = [-b_4, -b_1], \text{Ir}(B) = \emptyset$.

In the following, we will systematically investigate these sub-phases, solve the balance requirements for each case and subsequently obtain the corresponding BPEs.

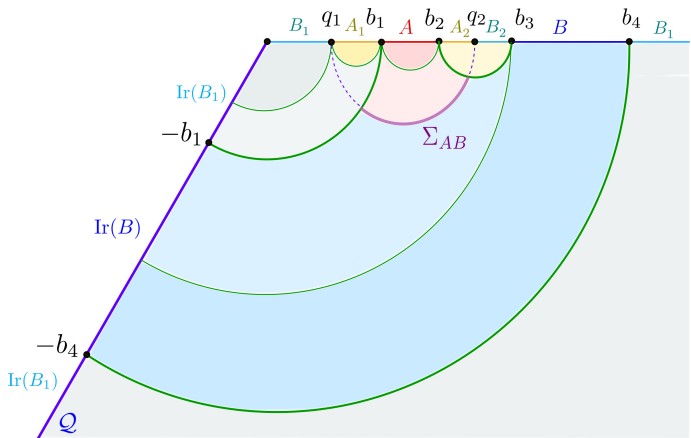

Figure 11: Phase-D2a: $\text{Ir}(A) = \emptyset$ and $\text{Ir}(B) \neq \emptyset$.

### 5.3.1 Phase-D2a

For $\text{Ir}(A) = \emptyset$ and $\text{Ir}(B) = [-b_4, -b_1]$, we assume that $AA_1A_2$ has no island, such that we have $\text{Ir}(A_1) = \text{Ir}(A_2) = \emptyset$, and the two balance points $q_1$ and $q_2$ lie at

$$0 < q_1 < b_1, \quad b_2 < q_2 < b_3. \tag{112}$$

The island region $(-\infty, 0]$ is divided into $\text{Ir}(B) = [-b_4, -b_1]$ and $\text{Ir}(B_1) = (-\infty, b_4) \cup (-b_1, 0]$. Furthermore we have $\text{Ir}(B_2) = \emptyset$ since $A_2B_2$ does not admit island. See Fig.11, for a schematics of the configuration.

The balance requirements in this case are the following two equations

$$\mathcal{I}(B\text{Ir}(B), A_1A_2A) = \mathcal{I}(A, B\text{Ir}(B)B_1\text{Ir}(B_1)B_2),$$
$$\mathcal{I}(B_2, A_1A_2A) = \mathcal{I}(A_2, B\text{Ir}(B)B_1\text{Ir}(B_1)B_2), \tag{113}$$

where the four PEEs in the above equations are calculated by:

$$
\begin{aligned}
\mathcal{I}(B\text{Ir}(B), A_1A_2A) &= \frac{1}{2}\Big[\tilde{S}_{B\text{Ir}(B)B_1\text{Ir}(B_1)} + \tilde{S}_{B\text{Ir}(B)B_2} - \tilde{S}_{B_2} - \tilde{S}_{B_1\text{Ir}(B_1)}\Big] \\
&= \frac{1}{2}\Big[\tilde{S}_{[q_1,b_3]} + \tilde{S}_{[-b_4,-b_1]\cup[q_2,b_4]} - \tilde{S}_{[q_2,b_3]} - \tilde{S}_{(\infty,-b_4)\cup[-b_1,q_1]\cup(b_4,\infty)}\Big] \\
&= \frac{c}{6}\log\frac{b_3 - q_1}{b_3 - q_2}\frac{q_2 + b_1}{q_1 + b_1},
\end{aligned}
$$

$$
\begin{aligned}
\mathcal{I}(A, B\text{Ir}(B)B_1\text{Ir}(B_1)B_2) &= \frac{1}{2}\Big[\tilde{S}_{AA_1} + \tilde{S}_{AA_2} - \tilde{S}_{A_1} - \tilde{S}_{A_2}\Big] \\
&= \frac{1}{2}\Big[\tilde{S}_{[q_1,b_2]} + \tilde{S}_{[b_1,q_2]} - \tilde{S}_{[q_1,b_1]} - \tilde{S}_{[b_2,q_2]}\Big] \\
&= \frac{c}{6}\log\frac{b_2 - q_1}{q_2 - b_2}\frac{q_2 - b_1}{b_1 - q_1},
\end{aligned}
$$

$$\tag{114}$$

and

$$\mathcal{I}(B_2, A_1 A_2 A) = \frac{1}{2}\Big[\tilde{S}_{B\mathrm{Ir}(B)B_1\mathrm{Ir}(B_1)B_2} + \tilde{S}_{B_2} - \tilde{S}_{B\mathrm{Ir}(B)B_1\mathrm{Ir}(B_1)}\Big]$$
$$= \frac{1}{2}\Big[\tilde{S}_{[q_1, q_2]} + \tilde{S}_{[q_2, b_3]} - \tilde{S}_{[q_1, b_3]}\Big]$$
$$= \frac{c}{6}\log\frac{(q_2 - q_1)(b_3 - q_2)}{\delta(b_3 - q_1)},$$

$$\mathcal{I}(A_2, B\mathrm{Ir}(B)B_1\mathrm{Ir}(B_1)B_2) = \frac{1}{2}\Big[\tilde{S}_{AA_1A_2} + \tilde{S}_{A_2} - \tilde{S}_{AA_1}\Big]$$
$$= \frac{1}{2}\Big[\tilde{S}_{[q_1, q_2]} + \tilde{S}_{[b_2, q_2]} - \tilde{S}_{[q_1, b_2]}\Big]$$
$$= \frac{c}{6}\log\frac{(q_2 - q_1)(q_2 - b_2)}{(b_2 - q_1)\delta}.$$

(115)

The solution to the two balance conditions are given by

$$q_1 = \frac{b_1^2 + b_2 b_3}{b_3 + b_2} - \frac{\sqrt{(b_2^2 - b_1^2)(b_3^2 - b_1^2)}}{b_3 + b_2}, \qquad q_2 = \frac{b_1^2 + b_2 b_3}{b_3 + b_2} + \frac{\sqrt{(b_2^2 - b_1^2)(b_3^2 - b_1^2)}}{b_3 + b_2}. \quad (116)$$

The corresponding BPE$(A : B)$ may be obtained as follows

$$\mathrm{BPE}(A : B) = \frac{c}{6}\log\frac{b_2 b_3 - b_1^2 + \sqrt{(b_2^2 - b_1^2)(b_3^2 - b_1^2)}}{b_1(b_3 - b_2)}, \quad (117)$$

which exactly matches with the area of EWCS in Phase D2a given in (60). Interestingly, in the holographic geometric picture, these two balance partition points are exactly located where the RT surface extending from the EWCS ends on the asymptotic boundary, as shown in Fig.11.

### 5.3.2 Phase-D2b

In this configuration, the partition point $x = -q$ divide the island region Is($AB$) into Ir($A$) $= [-q, -b_1]$ and Ir($B$) $= [-b_4, -q]$, as depicted in Fig.12. Also we have Ir($B_2$) $=$ Ir($A_2$) $= \emptyset$ since $A_2 B_2$ does not admit any island. Let us choose the other partition trivially as follows

$$A_1 = [0, b_1], \quad B_1 = [b_4, \infty]. \quad (118)$$

In this case, there are no ownerless island for $A_1$ and $B_1$. Thus we have Ir($A_1$) $= (-b_1, 0)$ and Ir($B_1$) $= [-\infty, -b_4)$. The balance requirements are given by the following two equations

$$\mathcal{I}(B\mathrm{Ir}(B), A\mathrm{Ir}(A)A_1\mathrm{Ir}(A_1)A_2) = \mathcal{I}(A\mathrm{Ir}(A), B\mathrm{Ir}(B)B_1\mathrm{Ir}(B_1)B_2),$$
$$\mathcal{I}(B_2, A\mathrm{Ir}(A)A_1\mathrm{Ir}(A_1)A_2) = \mathcal{I}(A_2, B\mathrm{Ir}(B)B_1\mathrm{Ir}(B_1)B_2),$$

(119)

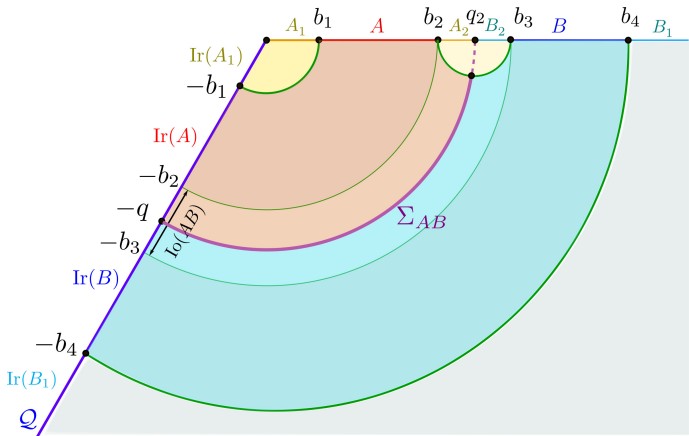

Figure 12: Phase-D2b: $\mathrm{Ir}(A) \neq \emptyset$ and $\mathrm{Ir}(B) \neq \emptyset$.

where the four PEEs in the above equations are calculated as follows

$$
\begin{aligned}
&\mathcal{I}(B\mathrm{Ir}(B), A\mathrm{Ir}(A)A_1\mathrm{Ir}(A_1)A_2)\\
&= \frac{1}{2}[\tilde{S}_{B\mathrm{Ir}(B)B_1\mathrm{Ir}(B_1)} + \tilde{S}_{B_2 B\mathrm{Ir}(B)} - \tilde{S}_{B_2} - \tilde{S}_{B_1\mathrm{Ir}(B_1)}]\\
&= \frac{1}{2}(\tilde{S}_{[-\infty,-q]\cup[b_3,+\infty]} + \tilde{S}_{[-b_4,-q]\cup[q_2,b_4]} - \tilde{S}_{[q_2,b_3]} - \tilde{S}_{[-\infty,-b_4]\cup[b_4,+\infty]})\\
&= \frac{c}{6}\log\frac{(b_3+q)(q+q_2)}{2q(b_3-q_2)} + \frac{c}{6}\kappa\,,
\end{aligned}
$$

(120)

$$
\begin{aligned}
&\mathcal{I}(A\mathrm{Ir}(A), B\mathrm{Ir}(B)B_1\mathrm{Ir}(B_1)B_2)\\
&= \frac{1}{2}[\tilde{S}_{A\mathrm{Ir}(A)A_1\mathrm{Ir}(A_1)} + \tilde{S}_{A_2 A\mathrm{Ir}(A)} - \tilde{S}_{A_1\mathrm{Ir}(A_1)} - \tilde{S}_{A_2}]\\
&= \frac{1}{2}(\tilde{S}_{[-q,b_2]} + \tilde{S}_{[-q,-b_1]\cup[b_1,q_2]} - \tilde{S}_{[-b_1,b_1]} - \tilde{S}_{[b_2,q_2]})\\
&= \frac{c}{6}\log\frac{(b_2+q)(q_2+q)}{2q(q_2-b_2)} + \frac{c}{6}\kappa\,,
\end{aligned}
$$

and

$$
\begin{aligned}
\mathcal{I}(B_2, A\mathrm{Ir}(A)A_1\mathrm{Ir}(A_1)A_2) &= \frac{1}{2}[\tilde{S}_{BB_1\mathrm{Ir}(BB_1)B_2} + \tilde{S}_{B_2} - \tilde{S}_{BB_1\mathrm{Ir}(BB_1)}]\\
&= \frac{1}{2}(\tilde{S}_{[-\infty,-q]\cup[q_2,\infty]} + \tilde{S}_{[q_2,b_3]} - \tilde{S}_{[-\infty,-q]\cup[b_3,+\infty]})\\
&= \frac{c}{6}\log\frac{(q_2+q)(b_3-q_2)}{\delta(b_3+q)}\,,
\end{aligned}
$$

$$
\begin{aligned}
\mathcal{I}(A_2, B\mathrm{Ir}(B)B_1\mathrm{Ir}(B_1)B_2) &= \frac{1}{2}[\tilde{S}_{A\mathrm{Ir}(A)A_1\mathrm{Ir}(A_1)A_2} + \tilde{S}_{A_2} - \tilde{S}_{A\mathrm{Ir}(A)A_1\mathrm{Ir}(A_1)}]\\
&= \frac{1}{2}(\tilde{S}_{[-q,q_2]} + \tilde{S}_{[b_2,q_2]} - \tilde{S}_{[-q,b_2]})\\
&= \frac{c}{6}\log\frac{(q_2+q)(q_2-b_2)}{\delta(b_2+q)}\,.
\end{aligned}
$$

(121)

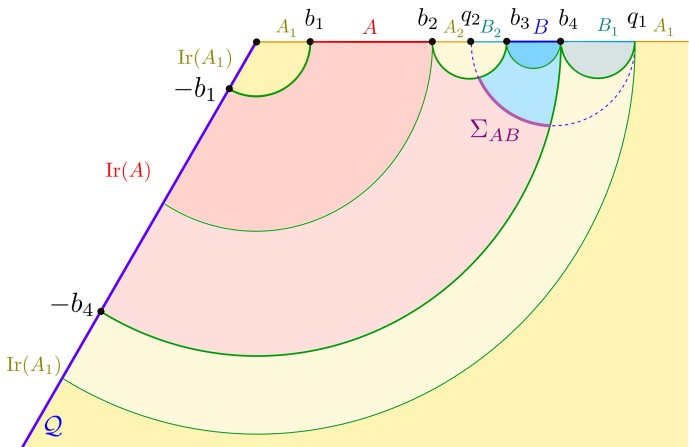

Figure 13: Phase-D2c: $\mathrm{Ir}(A) \neq \emptyset$ and $\mathrm{Ir}(B) = \emptyset$.

Similar to the adjacent cases, here the above two balance conditions coincide and are given by

$$\frac{b_3 + q}{b_3 - q_2} = \frac{b_2 + q}{q_2 - b_2}. \tag{122}$$

This means that there are infinite number of solutions to the balance requirements. Again, combining the balance requirements with the minimal requirement, the balance point should further satisfy the following extremal condition,

$$\partial_q[\mathcal{I}(B\mathrm{Ir}(B), A\mathrm{Ir}(A)A_1\mathrm{Ir}(A_1)A_2)] = \partial_q\left[\frac{c}{6}\log\frac{(b_3 + q)(q + q_2)}{2q(b_3 - q_2)} + \frac{c}{6}\kappa\right] = 0. \tag{123}$$

Solving these constraints, we arrive at

$$q = q_2 = \sqrt{b_2 b_3}. \tag{124}$$

Then the BPE for this phase is given by

$$\mathrm{BPE}(A:B) = \frac{c}{6}\log\frac{\sqrt{b_3} + \sqrt{b_2}}{\sqrt{b_3} - \sqrt{b_2}} + \frac{c}{6}\kappa, \tag{125}$$

which coincides with the area of EWCS in Phase D2b (62).

Similar to the Phase-A2b, we can transfer a portion, for example $[-q_1, q_1]$ of $A_1\mathrm{Ir}(A_1)$ to $B_1\mathrm{Ir}(B_1)$ as long as $A_1$ admit island. In such configurations the BPE is the same as the above result.

### 5.3.3 Phase-D2c

For $\mathrm{Ir}(A) = [-b_4, -b_1], \mathrm{Ir}(B) = \emptyset$, the configuration is symmetric to Phase-D2a in the exchange of $A$ and $B$ (see Fig.13). Following the same arguments as in phase-D2a, we arrive at two balance points

$$q_1 = \frac{b_4^2 + b_2 b_3}{b_3 + b_2} + \frac{\sqrt{(b_2^2 - b_4^2)(b_3^2 - b_4^2)}}{b_3 + b_2}, \qquad q_2 = \frac{b_4^2 + b_2 b_3}{b_3 + b_2} - \frac{\sqrt{(b_2^2 - b_4^2)(b_3^2 - b_4^2)}}{b_3 + b_2}, \tag{126}$$

and the corresponding BPE$(A:B)$ is given by

$$\mathrm{BPE}(A:B) = \frac{c}{6}\log\frac{b_2 b_3 - b_4^2 + \sqrt{(b_2^2 - b_4^2)(b_3^2 - b_4^2)}}{b_4(b_3 - b_2)}, \tag{127}$$

which matches with the area of the EWCS given in (64), for the Phase D2c.

### 5.3.4 Minimizing the BPE in Phase-D2

Now we compare the above three BPEs, which correspond to the three saddle EWCS, and choose the minimal one.

The critical point between phase-D2a and phase-D2b is given by

$$f(b_1, b_2, b_3) \equiv \frac{c}{6} \log \frac{\left(b_3 b_2 - b_1^2 + \sqrt{\left(b_3^2 - b_1^2\right)\left(b_2^2 - b_1^2\right)}\right)^2}{b_1^2 \left(\sqrt{b_3} + \sqrt{b_2}\right)^4} = \frac{c}{3}\kappa. \tag{128}$$

When $f < \frac{c}{3}\kappa$, phase-D2a gives the smaller BPE. Now we show that in this case, $A_1 A A_2$ does not admit an island. Let us compare the entanglement entropy for $A_1 A A_2$ in island and no-island saddles,

$$\begin{aligned}
S_{\text{no-island}}(A_1 A A_2) - S_{\text{island}}(A_1 A A_2) &= \frac{c}{6} \log \frac{(q_2 - q_1)^2}{4 q_1 q_2} - \frac{c}{3}\kappa \\
&= \frac{c}{6} \log \frac{\left(b_2^2 - b_1^2\right)\left(b_3^2 - b_1^2\right)}{b_1^2 (b_2 + b_3)^2} - \frac{c}{3}\kappa \\
&\equiv g(b_1, b_2, b_3) - \frac{c}{3}\kappa.
\end{aligned} \tag{129}$$

Note that the difference

$$g(b_1, b_2, b_3) - f(b_1, b_2, b_3) = \frac{c}{6} \log \frac{(b_3^2 - b_1^2)(b_2^2 - b_1^2)(\sqrt{b_2} + \sqrt{b_3})^4}{(b_2 + b_3)^2 \left(b_3 b_2 - b_1^2 + \sqrt{\left(b_3^2 - b_1^2\right)\left(b_2^2 - b_1^2\right)}\right)^2}, \tag{130}$$

increases as $b_2 \to b_3$ and thus we have

$$g(b_1, b_2, b_3) - f(b_1, b_2, b_3) < g(b_1, b_3, b_3) - f(b_1, b_3, b_3) = 0, \tag{131}$$

that is, $g(b_1, b_2, b_3)$ is always smaller than $f(b_1, b_2, b_3)$. Then we arrive at the condition

$$S_{\text{no-island}}(A_1 A A_2) - S_{\text{island}}(A_1 A A_2) < f(b_1, b_2, b_3) - \frac{c}{3}\kappa < 0, \tag{132}$$

which confirms our assumption that $A_1 A A_2$ does not admit an island.

Similarly, by comparing the BPEs between phase-D2b and phase-D2c, we draw the conclusion that phase-D2c gives the smaller BPE when

$$\frac{c}{6} \log \frac{\left(b_3 b_2 - b_4^2 + \sqrt{\left(b_3^2 - b_4^2\right)\left(b_2^2 - b_4^2\right)}\right)^2}{b_4^2 \left(\sqrt{b_3} + \sqrt{b_2}\right)^4} < \frac{c}{3}\kappa. \tag{133}$$

Following the same argument, we could confirm that $B_1 B B_2$ does not admit an island when phase-D2c gives the smaller BPE.

## 5.4 Disjoint $AB$ with disconnected entanglement wedge

In the island phase, the EWCS of the entanglement wedge $\mathcal{E}_{AB}$ disappears as the entanglement wedge becomes disconnected. This happens when the interval sandwiched between $A$ and $B$ admits its own island. This immediately indicates that $\mathcal{I}(A, B) = 0$. Nevertheless, this is not a sufficient condition for $\text{BPE}(A : B) = 0$ as the BPE is defined as the PEE $\mathcal{I}(A, B_1 B B_2)$ at the balance point. Since the BPE is non-negative, we can prove the correspondence between the BPE and EWCS in this case by finding a configuration where the BPE vanishes.

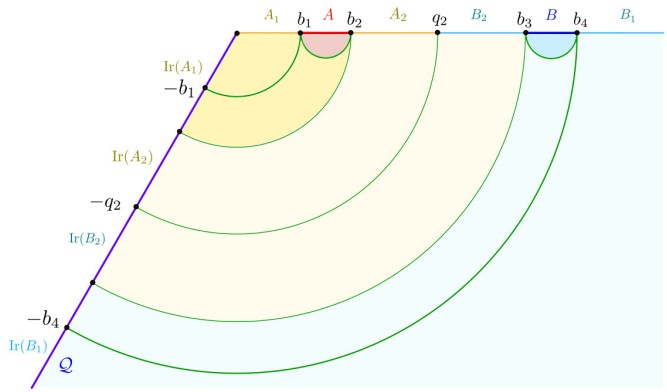

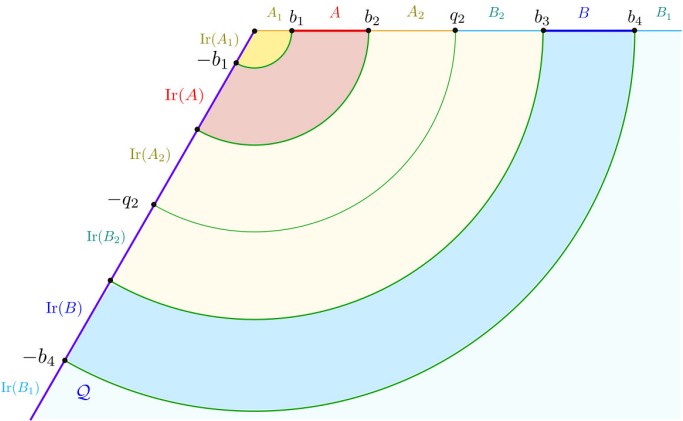

Figure 14: The disconnected phase for the entanglement wedge of $AB$. In this case, the interval $A_2B_2$ sandwiched between $A$ and $B$ admits an island. Top panel: when $A$ and $B$ do not have islands; Bottom panel: when $A$ and $B$ both have islands.

Let us consider the configuration depicted in Fig.14, where $A_2B_2$ admits island. In this case, according to our discussion in Sec.5.2.5 one can always find a partition point $q_2$ such that $A_2B_2$ is in Phase-A2b, hence we can choose

$$\text{Ir}(A_2) = [-q_2, -b_2], \quad \text{Ir}(B_2) = [-b_3, -q_2]. \tag{134}$$

Also in this case we have chosen

$$A_1 = [0, b_1], \quad B_1 = [b_4, \infty]. \tag{135}$$

When $A$ admits island we set

$$\text{Ir}(A) = \text{Is}(A) = [-b_2, -b_1], \quad \text{Ir}(A_1) = \text{Is}(A_1) = [-b_1, 0), \tag{136}$$

otherwise we set

$$\text{Ir}(A) = \emptyset, \quad \text{Ir}(A_1) = [-b_2, 0). \tag{137}$$

Also we apply a similar prescription to set $\text{Ir}(B)$ and $\text{Ir}(B_1)$.

According to our discussion for the Phase-A2b, the balance requirement

$$\mathcal{I}(A_2\text{Ir}(A_2), B_2BB_1\text{Ir}(B_1BB_2)) = \mathcal{I}(B_2\text{Ir}(B_2), A_2AA_1\text{Ir}(A_1AA_2)), \tag{138}$$

is always satisfied, and

$$\mathcal{I}(A_2 \mathrm{Ir}(A_2), B_2 B B_1 \mathrm{Ir}(B_1 B B_2)) = \mathrm{BPE}(A_2 : B_2) = \frac{c}{6} \log \frac{2q_2}{\delta} + \frac{c}{6}\kappa. \tag{139}$$

Then we test the other balance requirement

$$\mathcal{I}(A\mathrm{Ir}(A), B_2 B B_1 \mathrm{Ir}(B_1 B B_2)) = \mathcal{I}(B\mathrm{Ir}(B), A_2 A A_1 \mathrm{Ir}(A_1 A A_2)). \tag{140}$$

Let us first consider the cases where both of $A$ and $B$ do not admit island and hence $\mathrm{Ir}(A) = \mathrm{Ir}(B) = \emptyset$. We find that

$$
\begin{aligned}
\mathcal{I}(A, B_2 B B_1 \mathrm{Ir}(B_1 B B_2)) &= \frac{1}{2}\Big[\tilde{S}_{AA_2 \mathrm{Ir}(A_2)} + \tilde{S}_{A_1 \mathrm{Ir}(A_1) A_2} - \tilde{S}_{A_1 \mathrm{Ir}(A_1)} - \tilde{S}_{A_2 \mathrm{Ir}(A_2)}\Big] \\
&= \frac{1}{2}\left(\tilde{S}_{[-q_2, -b_2] \cup [b_1, q_2]} + \tilde{S}_{[-b_2, b_2]} - \tilde{S}_{[-b_2, b_1]} - \tilde{S}_{[-q_2, -b_2] \cup [b_2, q_2]}\right) \\
&= 0, \\
\mathcal{I}(B, A_2 A A_1 \mathrm{Ir}(A_1 A A_2)) &= \frac{1}{2}\Big[\tilde{S}_{BB_2 \mathrm{Ir}(B_2)} + \tilde{S}_{B_1 \mathrm{Ir}(B_1) B_2} - \tilde{S}_{B_1 \mathrm{Ir}(B_1)} - \tilde{S}_{B_2 \mathrm{Ir}(B_2)}\Big] \\
&= \frac{1}{2}\left(\tilde{S}_{[-b_3, -q_2] \cup [q_2, b_4]} + \tilde{S}_{[-b_3, b_3]} - \tilde{S}_{[-b_3, b_4]} - \tilde{S}_{[-b_3, -q_2] \cup [q_2, b_3]}\right) \\
&= 0,
\end{aligned}
\tag{141}
$$

where we have used the *basic proposal 2*. It is obvious that the second balance requirement is also satisfied. Hence the BPE between $A$ and $B$ vanishes,

$$\mathrm{BPE}(A : B) = 0. \tag{142}$$

Since the BPE should be non-negative, the above BPE is the minimal one. One can further check the cases where $A$ or $B$ admits island and get the same vanishing BPE. Then the vanishing BPE exactly matches to the vanishing EWCS.

## 5.5  BPE from minimizing the crossing PEE

The crossing PEE at the balance point has been shown to be minimal in vacuum CFTs [71]. Now we show that this also holds in the island phase. We pick phase-A2a and phase-D2a as examples.

For phase-A2a, the crossing PEEs are given by

$$
\begin{aligned}
\mathcal{I}(A, B_1 \mathrm{Ir}(B_1)) &= \frac{1}{2}\left(\tilde{S}_{AA_1} + \tilde{S}_{AB\mathrm{Ir}(B)} - \tilde{S}_{A_1} - \tilde{S}_{B\mathrm{Ir}(B)}\right) \\
&= \frac{1}{2}\left(\tilde{S}_{[q_1, b_2]} + \tilde{S}_{[-b_4, -b_1] \cup [b_1, b_4]} - \tilde{S}_{[q_1, b_1]} - \tilde{S}_{[-b_4, -b_1] \cup [b_2, b_4]}\right) \\
&= \frac{c}{6} \log\left[\frac{2b_1(b_2 - q_1)}{(b_1 + b_2)(b_1 - q_1)}\right],
\end{aligned}
\tag{143}
$$

and

$$
\begin{aligned}
\mathcal{I}(B\mathrm{Ir}(B), A_1) &= \frac{1}{2}\left(\tilde{S}_{B\mathrm{Ir}(B) \cup B_1 \mathrm{Ir}(B_1)} + \tilde{S}_{AB\mathrm{Ir}(B)} - \tilde{S}_A - \tilde{S}_{B_1 \mathrm{Ir}(B_1)}\right) \\
&= \frac{1}{2}\left(\tilde{S}_{(-\infty, q_1] \cup [b_2, \infty)} + \tilde{S}_{[-b_4, -b_1] \cup [b_1, b_4]} - \tilde{S}_{[b_1, b_2]} - \tilde{S}_{[-b_1, q_1] \cup (-\infty, -b_4] \cup [b_4, \infty)}\right) \\
&= \frac{c}{6} \log\left[\frac{2b_1(b_2 - q_1)}{(b_2 - b_1)(b_1 + q_1)}\right].
\end{aligned}
\tag{144}
$$

Then the total crossing PEE is

$$\frac{1}{2}\Big[\mathcal{I}(A, B_1 \mathrm{Ir}(B_1)) + \mathcal{I}(B\,\mathrm{Ir}(B), A_1)\Big] = \frac{c}{12}\log\left[\frac{4b_1^2(b_2 - q_1)^2}{(b_2^2 - b_1^2)(b_1^2 - q_1^2)}\right]. \qquad (145)$$

One easily finds that the extremal points $q_1$ for the crossing PEE are

$$q_1 = b_1^2/b_2, \quad \text{and} \quad q_1 = b_2\,. \qquad (146)$$

Since $0 < q_1 < b_1$, we have $q_1 = b_1^2/b_2$ as the point that minimizes the total crossing PEE. This is exactly the balanced point and the minimized total crossing PEE is given by

$$\Big[\mathcal{I}(A, B_1 \mathrm{Ir}(B_1)) + \mathcal{I}(B\,\mathrm{Ir}(B), A_1)\Big]\Big|_{minimal} = \frac{c}{3}\log 2\,, \qquad (147)$$

which may be identified as the lower bound of Markov gap $h(A : B)$ [109]. In non-island phase for adjacent intervals, the non-crossing PEE part in BPE exactly coincides with half the mutual information so that the crossing PEE part in BPE gives the Markov gap [71]. However, in island phase, the non-crossing PEE part in BPE is never equal to half the mutual information. For phase-A2a, we have the non-crossing PEE

$$\begin{aligned}
\mathcal{I}(A, B\,\mathrm{Ir}(B)) &= \frac{1}{2}\Big(\tilde{S}_A + \tilde{S}_{B_1 \mathrm{Ir}(B_1)A_1 A} - \tilde{S}_{B_1 \mathrm{Ir}(B_1)A_1}\Big) \\
&= \frac{1}{2}\Big(\tilde{S}_{[b_1, b_2]} + \tilde{S}_{[-\infty, -b_4] \cup [-b_1, b_2] \cup [b_4, \infty]} - \tilde{S}_{[-\infty, -b_4] \cup [-b_1, b_1] \cup [b_4, \infty]}\Big) \\
&= \frac{c}{6}\log\left(\frac{b_2 - b_1}{\delta}\frac{b_1 + b_2}{2b_1}\right),
\end{aligned} \qquad (148)$$

while half of the mutual information is given by

$$\begin{aligned}
\frac{1}{2}I(A : B) &= \frac{1}{2}(S_A + S_B - S_{AB}) \\
&= \frac{1}{2}(\tilde{S}_A + \tilde{S}_{B\,\mathrm{Is}(B)} - \tilde{S}_{AB\,\mathrm{Is}(AB)}) \\
&= \frac{c}{6}\log\left(\frac{b_2 - b_1}{\delta}\sqrt{\frac{b_2}{b_1}}\right).
\end{aligned} \qquad (149)$$

Thus the crossing PEE for phase-A2a should not exactly give the Markov gap.

Now we consider an example of disjoint phases, namely the phase-D2a. The crossing PEEs in this phase are given by

$$\begin{aligned}
\mathcal{I}(A, B_1 \mathrm{Ir}(B_1)) &= \frac{1}{2}\Big(\tilde{S}_{AA_1} + \tilde{S}_{AA_2 B_2 B \mathrm{Ir}(B)} - \tilde{S}_{A_1} - \tilde{S}_{A_2 B_2 B\,\mathrm{Ir}(B)}\Big) \\
&= \frac{1}{2}\Big(\tilde{S}_{[q_1, b_2]} + \tilde{S}_{[-b_4, -b_1] \cup [b_1, b_4]} - \tilde{S}_{[q_1, b_1]} - \tilde{S}_{[-b_4, -b_1] \cup [b_2, b_4]}\Big) \\
&= \frac{c}{6}\log\frac{2b_1(b_2 - q_1)}{(b_1 + b_2)(b_1 - q_1)}, \\
\mathcal{I}(A, B_2) &= \frac{1}{2}\Big(\tilde{S}_{AA_2} + \tilde{S}_{B\mathrm{Ir}(B)B_1 \mathrm{Ir}(B_1)A_1 A} - \tilde{S}_{A_2} - \tilde{S}_{B\mathrm{Ir}(B)B_1 \mathrm{Ir}(B_1)A_1}\Big) \\
&= \frac{1}{2}\Big(\tilde{S}_{[b_1, q_2]} + \tilde{S}_{[b_2, b_3]} - \tilde{S}_{[b_2, q_2]} - \tilde{S}_{[b_1, b_3]}\Big) \\
&= \frac{c}{6}\log\frac{(b_3 - b_2)(q_2 - b_1)}{(b_3 - b_1)(q_2 - b_2)},
\end{aligned} \qquad (150)$$

$$\mathcal{I}(B\mathrm{Ir}(B),A_1) = \frac{1}{2}\left(\tilde{S}_{AA_2B_2B\mathrm{Ir}(B)} + \tilde{S}_{B\mathrm{Ir}(B)B_1\mathrm{Ir}(B_1)} - \tilde{S}_{AA_2B_2} - \tilde{S}_{B_1\mathrm{Ir}(B_1)}\right)$$

$$= \frac{1}{2}\left(\tilde{S}_{[-b_4,-b_1]\cup[b_1,b_4]} + \tilde{S}_{[q_1,b_3]} - \tilde{S}_{[b_1,b_3]} - \tilde{S}_{[-\infty,-b_4]\cup[-b_1,q_1]\cup[b_4,\infty]}\right)$$

$$= \frac{c}{6}\log\frac{2b_1(b_3-q_1)}{(b_1+q_1)(b_3-b_1)},$$

$$\mathcal{I}(B\mathrm{Ir}(B),A_2) = \frac{1}{2}\left(\tilde{S}_{B_2B\mathrm{Ir}(B)} + \tilde{S}_{B\mathrm{Ir}(B)B_1\mathrm{Ir}(B_1)A_1A} - \tilde{S}_{B_2} - \tilde{S}_{B_1\mathrm{Ir}(B_1)A_1A}\right)$$

$$= \frac{1}{2}\left(\tilde{S}_{[-b_4,-b_1]\cup[q_2,b_4]} + \tilde{S}_{[b_2,b_3]} - \tilde{S}_{[q_2,b_3]} - \tilde{S}_{[-b_4,-b_1]\cup[b_2,b_4]}\right)$$

$$= \frac{c}{6}\log\frac{(q_2+b_1)(b_3-b_2)}{(b_1+b_2)(b_3-q_2)}.$$

Then the total crossing PEE is

$$\mathcal{I}(A,B_1\mathrm{Ir}(B_1)) + \mathcal{I}(A,B_2) + \mathcal{I}(B\mathrm{Ir}(B),A_1) + \mathcal{I}(B\mathrm{Ir}(B),A_2)$$
$$= \frac{c}{6}\log\left[\frac{4b_1^2(b_3-b_2)^2(q_2^2-b_1^2)(b_2-q_1)(b_3-q_1)}{(b_2+b_1)^2(b_3-b_1)^2(b_1^2-q_1^2)(q_2-b_2)(b_3-q_2)}\right]. \tag{151}$$

Again, one may find that the total crossing PEE is minimized at the points

$$q_1 = \frac{b_1^2+b_2b_3}{b_3+b_2} - \frac{\sqrt{(b_2^2-b_1^2)(b_3^2-b_1^2)}}{b_3+b_2}, \qquad q_2 = \frac{b_1^2+b_2b_3}{b_3+b_2} + \frac{\sqrt{(b_2^2-b_1^2)(b_3^2-b_1^2)}}{b_3+b_2}, \tag{152}$$

which are exactly the balance points.

# 6 Partial entanglement entropy and its geometric picture in island phase

In this section, we calculate the contributions to various PEEs via the generalized ALC formula under the assignment of the ownerless island that gives the minimal BPE. We will see that the contributions $s_{AB}(A)$ and $s_{AB}(B)$ correspond to the two portions of the RT surface of $AB$, which are divided by the point at which the EWCS $\Sigma_{AB}$ anchors on the RT surface $\mathcal{E}_{AB}$.

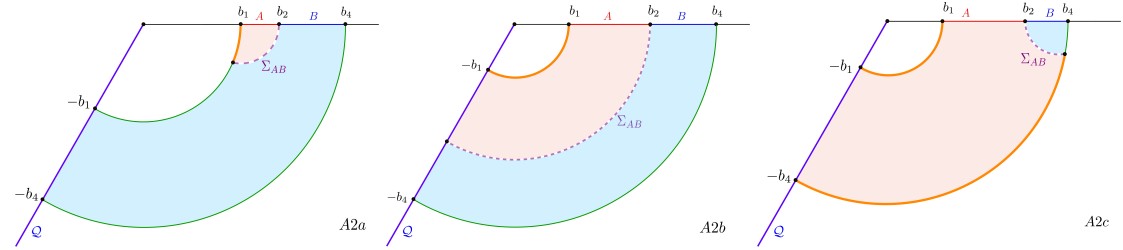

Figure 15: The correspondence between the geodesic chords (orange lines) on RT($A\cup B$) and the PEE $A$ to $AB$ for phase-A2.

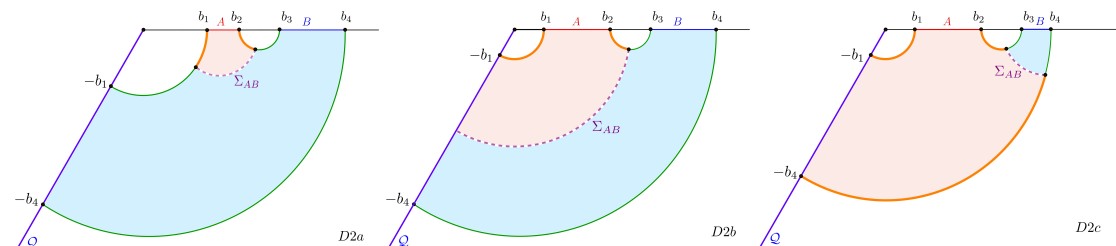

Figure 16: The correspondence between the geodesic chords (orange lines) on RT($A \cup B$) and the PEE $A$ to $AB$ for phase-D2.

## 6.1 Adjacent $AB$ with island

When BPE for phase-A2a minimizes, we have $\mathrm{Ir}(A) = \emptyset$ and $\mathrm{Ir}(B) = [-b_4, -b_1]$ and

$$
\begin{aligned}
s_{AB}(A) &= \frac{1}{2}(\tilde{S}_{A\,\mathrm{Ir}(A)\,B\,\mathrm{Ir}(B)} + \tilde{S}_A - \tilde{S}_{B\,\mathrm{Ir}(B)}) \\
&= \frac{1}{2}(\tilde{S}_{[-b_4,-b_1]\cup[b_1,b_4]} + \tilde{S}_{[b_1,b_2]} - \tilde{S}_{[-b_4,-b_1]\cup[b_2,b_4]}) \\
&= \frac{c}{6}\log\frac{2b_1(b_2-b_1)}{\delta(b_2+b_1)}.
\end{aligned}
\tag{153}
$$

This is just the area of the geodesic chord (see the orange line in Fig.15) on RT($b_1$), which is determined by the EWCS $\Sigma_{AB}$.

When BPE for phase-A2b minimizes, we have $\mathrm{Ir}(A) = [-b_2, -b_1], \mathrm{Ir}(B) = [-b_4, -b_2]$ and

$$
\begin{aligned}
s_{AB}(A) &= \frac{1}{2}(\tilde{S}_{A\,\mathrm{Ir}(A)\,B\,\mathrm{Ir}(B)} + \tilde{S}_{A\,\mathrm{Ir}(A)} - \tilde{S}_{B\,\mathrm{Ir}(B)}) \\
&= \frac{1}{2}(\tilde{S}_{[-b_4,-b_1]\cup[b_1,b_4]} + \tilde{S}_{[-b_2,-b_1]\cup[b_1,b_2]} - \tilde{S}_{[-b_4,-b_2]\cup[b_2,b_4]}) \\
&= \frac{c}{6}\log\frac{2b_1}{\delta} + \frac{c}{6}\kappa,
\end{aligned}
\tag{154}
$$

which is just the area of RT($b_1$).

When BPE for phase-A2c minimizes, we have $\mathrm{Ir}(A) = [-b_4, -b_1], \mathrm{Ir}(B) = \emptyset$ and

$$
\begin{aligned}
s_{AB}(A) &= \frac{1}{2}(\tilde{S}_{A\,\mathrm{Ir}(A)\,B\,\mathrm{Ir}(B)} + \tilde{S}_{A\,\mathrm{Ir}(A)} - \tilde{S}_B) \\
&= \frac{1}{2}(\tilde{S}_{[-b_4,-b_1]\cup[b_1,b_4]} + \tilde{S}_{[-b_4,-b_1]\cup[b_1,b_2]} - \tilde{S}_{[b_2,b_4]}) \\
&= \left(\frac{c}{6}\log\frac{2b_1}{\delta} + \frac{c}{6}\kappa\right) + \left(\frac{c}{6}\log\frac{b_4+b_2}{b_4-b_2} + \frac{c}{6}\kappa\right),
\end{aligned}
\tag{155}
$$

where the first term is the area of RT($b_1$) and the second term is the area of the geodesic chord (see the orange lines in Fig.15) on RT($b_4$), which is determined by the EWCS $\Sigma_{AB}$.

## 6.2 Disjoint $AB$ with island

When BPE for phase-D2a minimizes, we have $\mathrm{Ir}(A) = \emptyset$ and $\mathrm{Ir}(B) = [-b_4, -b_1]$ and

$$
\begin{aligned}
s_{AB}(A) &= \frac{1}{2}(\tilde{S}_{A\,\mathrm{Ir}(A)\,B\,\mathrm{Ir}(B)} + \tilde{S}_A - \tilde{S}_{B\,\mathrm{Ir}(B)}) \\
&= \frac{1}{2}(\tilde{S}_{[-b_4,-b_1]\cup[b_1,b_2]\cup[b_3,b_4]} + \tilde{S}_{[b_1,b_2]} - \tilde{S}_{[-b_4,-b_1]\cup[b_3,b_4]}) \\
&= \frac{c}{6}\log\frac{2b_1(b_3-b_2)(b_2-b_1)}{\delta^2(b_3+b_1)},
\end{aligned}
\tag{156}
$$

which can be further rewritten as

$$s_{AB}(A) = \frac{c}{6}\log\frac{2b_1(q_2 - b_1)}{(b_1 + q_2)\delta} + \frac{c}{6}\log\frac{(b_3 - b_2)(q_2 - b_2)}{(b_3 - q_2)\delta}, \tag{157}$$

with the balance point $q_2$ given by

$$q_2 = \frac{b_1^2 + b_2 b_3 + \sqrt{(b_2^2 - b_1^2)(b_3^2 - b_1^2)}}{b_3 + b_2}. \tag{158}$$

The two terms in eq.(157) are just the areas of the geodesic chords (see the orange lines in the left panel of Fig.16) on $RT(b_1)$ and the RT surface of the sandwiched interval $[b_2, b_3]$, respectively and they are determined by the EWCS $\Sigma_{AB}$.

When BPE for phase-D2b minimizes, we have $\text{Ir}(A) = [-q_2, -b_1], \text{Ir}(B) = [-b_4, -q_2]$ and

$$\begin{aligned}
s_{AB}(A) &= \frac{1}{2}(\tilde{S}_{A\,\text{Ir}(A)\,B\,\text{Ir}(B)} + \tilde{S}_{A\,\text{Ir}(A)} - \tilde{S}_{B\,\text{Ir}(B)}) \\
&= \frac{1}{2}(\tilde{S}_{[-b_4,-b_1]\cup[b_1,b_2]\cup[b_3,b_4]} + \tilde{S}_{[-q_2,-b_1]\cup[b_1,b_2]} - \tilde{S}_{[-b_4,-q_2]\cup[b_3,b_4]}) \\
&= \left(\frac{c}{6}\log\frac{2b_1}{\delta} + \frac{c}{6}\kappa\right) + \frac{c}{6}\log\frac{(b_3 - b_2)(b_2 + q_2)}{(b_3 + q_2)\delta},
\end{aligned} \tag{159}$$

where $q_2 = \sqrt{b_2 b_3}$. The first term is just the area of $RT(b_1)$ and the second term is the area of the geodesic chord (see the orange lines in the middle panel of Fig.16) on the RT surface associated with the sandwiched interval $[b_2, b_3]$, which is also determined by the EWCS $\Sigma_{AB}$.

When BPE for phase-D2c minimizes, we have $\text{Ir}(A) = [-b_4, -b_1], \text{Ir}(B) = \emptyset$ and

$$\begin{aligned}
s_{AB}(A) &= \frac{1}{2}(\tilde{S}_{A\,\text{Ir}(A)\,B\,\text{Ir}(B)} + \tilde{S}_{A\,\text{Ir}(A)} - \tilde{S}_B) \\
&= \frac{1}{2}(\tilde{S}_{[-b_4,-b_1]\cup[b_1,b_2]\cup[b_3,b_4]} + \tilde{S}_{[-b_4,-b_1]\cup[b_1,b_2]} - \tilde{S}_{[b_3,b_4]}) \\
&= \frac{c}{6}\log\frac{2b_1(b_3 - b_2)(b_2 + b_4)}{\delta^2(b_4 - b_3)} + \frac{c}{3}\kappa,
\end{aligned} \tag{160}$$

which can be further written as

$$s_{AB}(A) = \left(\frac{c}{6}\log\frac{2b_1}{\delta} + \frac{c}{6}\kappa\right) + \left(\frac{c}{6}\log\left(\frac{q_2 + b_4}{b_4 - q_2}\right) + \frac{c}{6}\kappa\right) + \frac{c}{6}\log\frac{(b_3 - b_2)(q_2 - b_2)}{(b_3 - q_2)\delta}, \tag{161}$$

with the balance point given by

$$q_2 = \frac{b_4^2 + b_2 b_3 - \sqrt{(b_2^2 - b_4^2)(b_3^2 - b_4^2)}}{b_3 + b_2}. \tag{162}$$

The first term in eq.(161) is just the area of $RT(b_1)$, the second and the third term in (161) are the areas of the geodesic chords (see the orange lines in the right panel of Fig.16) on $RT(b_4)$ and the RT surface of the sandwiched interval $[b_2, b_3]$, respectively and they are determined by the EWCS $\Sigma_{AB}$.

# 7 Discussion

## 7.1 Summary

In this paper, we have explored the entanglement structure in the island phase in the context of partial entanglement entropy. Despite several primitive attempts [80,83,101], this remains

quite an unexplored aspect of entanglement islands. Based on the claim [49] that a system in island phase has self-encoding property, we conclude that calculating the entanglement entropy of a region $A$ involves the degrees of freedom in the island Is($A$), which is outside this region. This property essentially changes the way we evaluate the contribution to entanglement entropy $S_A$ from subsets in $A$. Firstly, the island Is($A$) should be understood as a window through which $A$ can entangle with degrees of freedom outside $A \cup$ Is($A$). Secondly, when we consider the contribution from a subregion $\alpha$, we should also include the contribution from Is($\alpha$), or the generalized (or reflected) island Ir($\alpha$) if there are ownerless island regions. With the island contributions taken into account, we find a generalized version of the ALC proposal to construct the PEE and a generalized version of the balance requirement to define the BPE in the island phase.

For configurations without ownerless islands, the assignments of the island regions to the subsets is clear with no ambiguity. Nevertheless, in configurations with ownerless island regions, there is no intrinsic rule to clarify these assignments. For any choice of the assignment, we can solve the generalized balance requirements and calculate the BPE. Remarkably, we find that the BPEs for different assignments of the ownerless island correspond to different saddles of the EWCS. Then it is natural to choose the assignment that gives the minimal BPE. Furthermore, with the assignment of ownerless island settled, we calculate the contributions $s_{AB}(A)$ and $s_{AB}(B)$ and explore their geometric picture, which is consistent with the geometric picture in non-island phase.

We stress that, for the gravitational Set-up 1 and the Set-up 2, the following three proposals for the island phases are enough to get our results.

1. As in the no-island phases, the PEE structure of the island phase is also described by the two-point PEEs $\mathcal{I}(x, y)$ which is unaffected by how we divide the system.

2. When we consider any type of correlation between two spacelike separated regions $A$ and $B$, we should also properly take into account the contributions from their island regions as well as from the ownerless islands.

3. The two-point functions of the twist operators for non-symmetric intervals in 2d effective field theories are well defined and they represent the PEE following the *basic proposal 1*.

The physical meaning of this paper is multifaceted. On one hand, our results give a nontrivial test to the correspondence between the BPE and the EWCS, and to the purification independence of the BPE. On the other hand, they indicate that the above listed proposals are highly consistent. These proposals give a finer description for the entanglement structure of the island phases. Testing and proving the above proposals or conjectures from other perspectives will be important future directions.

Using the PEE structure to study unitary evolution of gravitational theories, especially the black hole evaporation, will be quite interesting. This has been partially explored in [80, 83], where they calculated the entanglement contour function for the radiation region following the ALC proposal[11] and find that there are vanishing PEEs for certain regions. According to [80] this is a reflection of the protection of bulk island regions against erasures of the boundary state. It will be very interesting to take a deeper look at this problem in the future.

## 7.2 More on the self-encoding property

Actually, the self-encoding property is not necessary in the gravitational set-ups if one directly starts from the above three proposals. It is necessary when we consider the holographic Weyl

---

[11]As was pointed out by our results, the ALC proposal should be modified in island phases. So the results in [80, 83] need further consideration.

transformed CFT$_2$ of Set-up 1 which is non-gravitational, where we should assume that the self-encoding property emerges if the island formula $I$ gives smaller entanglement entropy in this non-gravitational toy model [49]. Nevertheless, the self-encoding property gives us guidelines to arrive at the above three proposals. It tells us how to explicitly deal with the contribution from the island regions when computing the PEE and BPE. More importantly, it tells us that the two-point functions of the twist operators in the Weyl transformed CFT$_2$ do not always give us well-defined entanglement entropy, which help us get to the *basic proposal 1*.

The self-encoding property is essentially the property that, certain space-like separated degrees of freedom are not independent from each other. The dependence between spacelike separated degrees of freedom is quite special as it goes beyond causality. This property is emergent from certain constraints to the whole system, which have highly non-local effects. The corresponding constraints are not only imposed on a single state, but on the Hilbert space. In other words, the constraints should make sure that any states in the reduced Hilbert space should be confined in the reduced space under evolution. Explicit construction of such constraints in any toy models is highly non-trivial and a very interesting topic to explore. This may lead us to the realization of entanglement islands in non-gravitational systems that can be prepared in the lab.

In gravitational systems with entanglement islands the self-encoding property is nothing but the statement that, the state of the entanglement island can be reconstructed from the Hawking radiation. The coding relation involves two different subsets of one system. This is different from the reconstruction of the entanglement wedge from the boundary dual subregion in holography, where the coding relation involves two subsystems on different sides of the duality. As was pointed out in [10, 11], the derivation of the island formula based on the replica wormhole configurations does not rely on the existence of holography. The self-encoding property has a holographic description only in the doubly holography set-ups, where the island region is considered to be part of the entanglement wedge of the other region.

Besides the original discussion in [49], there are many extremely interesting future directions that deserve further exploration regarding the self-encoding property in gravitational systems. For example, what is the coding relation that lead us to the island formula $I$ in gravitational systems? And what are the constraints in gravitational systems that vastly reduce the Hilbert space, hence lead to the corresponding coding relation? Is the cutoff-dependent Weyl transformation that wipes out the UV physics a good simulation for the reduction of the Hilbert space in gravitational theories? It will be very interesting to consider other configurations of the Weyl transformation which result in different bulk cutoff branes where entanglement islands emerge. We can study the entanglement islands, EWCS and BPE in such scenarios, and compare with the other generalized configurations of AdS/BCFT, like the holographic BCFT with two boundaries [12], the wedge holography [110–112], and the bulk brane with perturbations [113–115].

## 7.3   The BPE and the reflected entropy

The BPE and EWCS are closely related to the reflected entropy. In holographic setups, evidence is given in [55] for the correspondence between the EWCS and the reflected entropy. Also in [70] it was shown that the BPE in the canonical purification reduces to the reflected entropy as the reflection symmetry in the canonical purification automatically satisfies the balance requirements. In other words, the reflection symmetry in the canonical purification is just the balance requirements. Nevertheless, searching for the EWCS is an optimization problem which is quite different from the balance requirements. Hence, it is reasonable to believe that the EWCS is dual to some quantum information quantity which is defined under optimization, rather than the reflected entropy or BPE. Moreover, an explicit example has been provided in [116], where the reflected entropy is not monotonically decreasing under partial trace. This

indicates that the reflected entropy is not a physical measure of mixed-state correlations. Given the proposal that the BPE is a generalization of the reflected entropy to generic purifications, this criticism also applies to the BPE.

Here we give some clarification on why the BPE is related to an optimization problem, and why the study on the BPE and the reflected entropy is still important. In [55] the double copy of an entanglement wedge glued along the RT surface is given as a case of canonical purification in holography. Let us consider a generic mixed state $AB$ and its canonical purification $ABA_1B_1$. In these configurations the balance requirement, i.e. the reflection symmetry with respect to the RT surface $\mathcal{E}_{AB}$ of $AB$, requires $\mathcal{E}_{AA_1}$ to be normal to $\mathcal{E}_{AB}$. Such a normal relation is usually a necessary condition for the solution of an optimization problem. Then it is tempting to believe that, the solutions to the balance requirements beyond the canonical purification configurations also characterize the saddle points of this optimization problem. If we find all the saddle points, we should choose the one that gives the minimized value. In other words we propose that, solving the balance requirements happens to solve the optimization problem in many configurations. In this paper we encounter many examples where the number of the solutions to the balance requirements are multiple or even infinite; they correspond to different saddle points of the EWCS. And the minimal BPE exactly matches with the minimal EWCS. These observations give us clues to relate the BPE to an optimization problem, and a through demonstration of this statement will an important future direction.

A more important clue comes from [71,72] and Sec.5.5 in this paper, where the authors find that the minimization of the crossing PEE, which is a pure optimization problem, coincides with the computation of BPE, as well as the EWCS. This observation has passed several non-trivial test, hence indicates that the minimized crossing PEE would be a more appropriate quantum information quantity that corresponds to the EWCS in holography. Also the minimized crossing PEE can be defined in non-holographic configurations. It is important to address that the minimization of the crossing PEE is a priori independent of the balance requirements, hence it is possible that the BPE and the minimized crossing PEE are not equivalent in general. It is also possible that the minimized crossing PEE is monotonically decreasing under partial trace in general while the reflected entropy or BPE is not. We leave this for future investigations.

# Acknowledgments

We would like to thank Rongxin Miao, Tadashi Takayanagi, Huajia Wang and Yang Zhou for helpful discussion. We also thank the referees of Scipost Physics for giving useful suggestions to clarify the set-ups and the role of the self-encoding property in this work.

Q.Wen would like to thank the Institute of Theoretical Physics, Chinese Academy of Sciences and the Yanqi Lake Beijing Institute of Mathematical Sciences and Applications for kind hospitality during the development of this project.

**Funding information** J.Lin is supported by the National Natural Science Foundation of China under Grant No.12247117, No.12247103 and No.12047502. Y.Lu is supported by the China Postdoctoral Science Foundation under Grant No.2022TQ0140 and the National Natural Science Foundation of China under Grant No.12247161. Q.Wen thank the "Zhishan Scholars" program of Southeast University for support.

# A Setup2: A generalized version of AdS/BCFT

In the appendix, we study the BPE in the setup of a generalized model of $AdS_3/BCFT_2$ [9, 97–99, 104], where the theory on the left-hand-side $x < 0$ couples to gravity, so we can directly apply the island formula (2). However, in the standard $AdS_3/BCFT_2$ correspondence [12] only symmetric two-point functions for the twist operators are well-defined in the effective description. The purpose of our generalization is to allow the evaluation of the non-symmetric two-point functions for twist operators. This maybe achieved by adding additional matter fields on the EoW brane such that, the theory on the brane is no longer just the gravity dual of the boundary point of the BCFT. It maybe more appropriate to describe the 2d effective theory picture as a gravitational $CFT_2$ coupled to a bath $CFT_2$ with a transparent boundary condition. In the following we will focus on a specific model namely the defect extremal surface (DES) model proposed in [97].

## A.1 A brief review of $AdS_3/BCFT_2$ and the DES model

The holographic dual of a $CFT_2$ with a boundary (BCFT) is proposed to be an $AdS_3$ spacetime with a co-dimension one brane where Neumann boundary condition is imposed [12]. The action of the bulk spacetime is given by

$$I_{\text{AdS}} = \frac{1}{16\pi G_N} \int_{\mathcal{N}} \mathrm{d}^3 x \sqrt{-g}(R - 2\Lambda) + \frac{1}{8\pi G_N} \int_{\mathcal{Q}} \mathrm{d}^2 x \sqrt{-h}(K - T), \qquad (A.1)$$

where $\mathcal{N}$ and $\mathcal{Q}$ denotes the bulk and the brane, respectively. $T$ is the tension of the brane, which determines its location in the bulk. Working in the coordinates

$$\mathrm{d}s^2 = \mathrm{d}\rho^2 + \ell^2 \cosh^2 \frac{\rho}{\ell} \frac{-\mathrm{d}t^2 + \mathrm{d}w^2}{w^2} \qquad (A.2)$$

$$= \frac{\ell^2}{z^2} \left( -\mathrm{d}t^2 + \mathrm{d}y^2 + \mathrm{d}z^2 \right), \qquad (A.3)$$

where $\ell$ is the radius of $AdS_3$, the brane is settled at

$$\rho = \rho_0, \qquad (A.4)$$

where $\rho_0$ is determined by the tension $T = \frac{1}{\ell} \tanh \frac{\rho_0}{\ell}$. Note that, in the Set-up 1 the cutoff brane in the bulk is settled at $\rho = \kappa$ [49], which means the two setups coincide provided $\kappa = \rho_0$. The polar coordinate $\theta$ is related to $\rho$ via $(\cos\theta)^{-1} = \cosh(\rho/\ell)$. We see that the brane is orthogonal to the boundary, namely $\theta(\rho_0) = 0$, iff $T = 0$. Turning on the tension or adding other matter on the brane moves $\theta(\rho_0)$ away from zero. For an interval $A = [0, L]$ containing the boundary of BCFT, the entanglement entropy of $A$ is also calculated by the RT formula [12]

$$S_A = \frac{\text{Area}[\gamma_A]}{4G_N} = \frac{c}{6} \log \frac{2L}{\delta} + \frac{c}{6} \text{arctanh}(\sin\theta_0), \qquad (A.5)$$

where the second term $\frac{c}{6}\text{arctanh}(\sin\theta_0)$ is the boundary entropy for BCFT.

The DES model proposed in [97] considers conformal matters (for example a defect theory) living on the brane, and correspondingly the action has an additional term for the conformal matter,

$$I_{\text{AdS}} = \frac{1}{16\pi G_N} \int_{\mathcal{N}} \mathrm{d}^3 x \sqrt{-g}(R - 2\Lambda) + \frac{1}{8\pi G_N} \int_{\mathcal{Q}} \mathrm{d}^2 x \sqrt{-h}(K - T) + I_{\text{CFT},\mathcal{Q}}. \qquad (A.6)$$

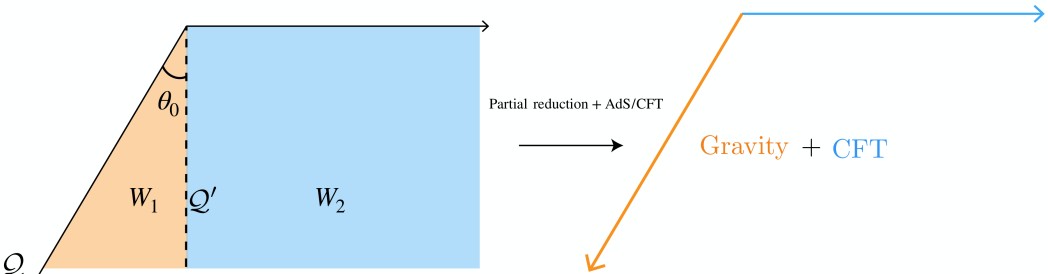

Figure 17: This figure is extracted from [109]. 2d effective theory description for DES model through the partial reduction for $W_1$ and holographic duality for $W_2$.

The field equation or equivalently the Neumann boundary condition on the brane is

$$K_{ab} - (K - T)h_{ab} = 8\pi G_N \chi h_{ab}, \tag{A.7}$$

where $h_{ab}$ is the induced metric on the brane. In the above equation $\chi$ characterizes the CFT matter and is related to its central charge $c'$. Then the central charge $c'$ for matters on the brane is determined by solving (A.7) and

$$c' = 2\cosh^2 \frac{\rho_0}{\ell}\left(\tanh\frac{\rho_0}{\ell} - \ell T\right)c, \tag{A.8}$$

where $c = \frac{3\ell}{2G_N}$ is the central charge of the CFT dual to the bulk $W_2$ wedge. In the following, we choose the tension $T$ such that $c' = c$ as in [97].

The 2d effective description for the DES model can be obtained by partial dimensional reduction. Let us decompose the bulk into two parts $W_1$ and $W_2$ along the (imaginary) surface $Q'$ with $(t, x, y) = (t, 0, y)$ (see Fig.17 for illustration). Then, we perform the Randall-Sundrum reduction along $\rho$ direction on $W_1$ to obtain a 2d topological gravity theory plus CFT matter living on the brane. The gravity on the brane is purely induced from the reduction of the $W_1$ bulk region. The $CFT_2$ bath is obtained as the dual to the $W_2$ wedge. Ultimately, one arrives at the 2d effective description for DES model, a 2d topological gravity + CFT defect matter living on the brane coupled with flat CFT bath along the $x = 0$ surface.

From the $AdS_3$ bulk perspective, according to the RT-like DES proposal [97], the entanglement entropy for a BCFT interval $A = [0, L]$ is given by the area of the RT surface $\Gamma$, which connects the boundary point $x = L$ and the brane, plus the defect term

$$
\begin{aligned}
S_A &= \frac{\text{Area}(\Gamma)}{4G_N} + S_{\text{defect}} \\
&= \frac{c}{6}\log\frac{2L}{\delta} + \frac{c}{6}\text{arctanh}(\sin\theta_0) + \frac{c}{6}\log\left(\frac{2\ell}{\delta_y \cos\theta_0}\right),
\end{aligned} \tag{A.9}
$$

where $\delta_y$ is the UV regulator on the brane. The defect term is calculate by the one-point function of the twist operator on the brane,

$$S_{\text{defect}} = \frac{c}{6}\log\left(\frac{2\ell}{\delta_y \cos\theta_0}\right). \tag{A.10}$$

It should be understood as the bulk semi-classical entanglement entropy in the RT formula with quantum correction [6, 105].

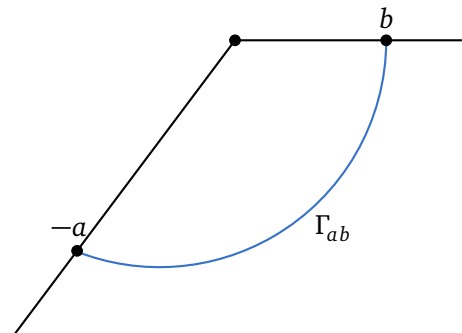

Figure 18: The blue curve denotes the geodesic $\Gamma_{ab}$ connecting the point at the boundary $x = b$ and the point $y = -a$ on the brane in DES setup.

From the 2-dimensional effective field theory perspective, we can apply the island formula to calculate the entanglement entropy. For example, the entanglement entropy for an interval $[0, L]$ can be calculated by minimizing the generalized entropy

$$S_{gen}([-a, L]) = S_{\text{area}}(-a) + \tilde{S}_{[-a, L]}, \tag{A.11}$$

where $\tilde{S}_{[y_1, y_2]}$ is the effective entropy of CFT matters in this 2d perspective [8, 11],

$$\tilde{S}_{[y_1, y_2]} = \frac{c}{6} \log \left( \frac{|y_1 - y_2|^2}{\delta_{y_1} \delta_{y_2} \Omega(y_1, \bar{y}_1) \Omega(y_2, \bar{y}_2)} \right), \quad ds^2 = \Omega^{-2} dy \, d\bar{y}, \tag{A.12}$$

with

$$\Omega(y) = \begin{cases} \left| \frac{y \cos \theta_0}{\ell} \right|, & \text{for } y < 0 \text{ on the brane}, \\ 1, & \text{for } y > 0 \text{ on the flat CFT}. \end{cases} \tag{A.13}$$

It is easy to see that the formula (A.12) is also a Weyl transformed two-point function[12] of the CFT, similar to (17). The area term is given by [97]

$$S_{\text{area}} = \frac{1}{4G_N^{\text{brane}}} = \frac{\rho_0}{4G_N} = \frac{c}{6} \operatorname{arctanh}(\sin \theta_0). \tag{A.14}$$

It is easy to find that, the generalized entropy is minimized at $a = L$, and the result exactly agrees with eq.(A.9).

## A.2 Calculations of BPE

Let us first calculate the PEE in the 2d effective description of the DES model. For a connected interval $\gamma = [-a, b]$ where $a, b > 0$ and the points $x < 0$ lives on the brane, the PEE is given by

$$\begin{aligned} \mathcal{I}(\gamma, \bar{\gamma}) &= S_{gen}([-a, b]) = S_{\text{area}}(-a) + \tilde{S}_{[-a, b]} \\ &= \frac{c}{6} \operatorname{arctanh}(\sin \theta_0) + \frac{c}{6} \log \frac{(b + a)^2 l}{a \cos \theta_0 \delta \delta_y}. \end{aligned} \tag{A.15}$$

When $a = b$, eq.(A.15) becomes

$$\mathcal{I}(\gamma, \bar{\gamma}) = \frac{c}{6} \log \frac{2b}{\delta} + \frac{c}{6} \operatorname{arctanh}(\sin \theta_0) + \frac{c}{6} \log \frac{2l}{\delta_y \cos \theta_0}. \tag{A.16}$$

---

[12]Note that, the Weyl factor $\Omega(y)$ in this case is also non-smooth at the interface $y = 0$.

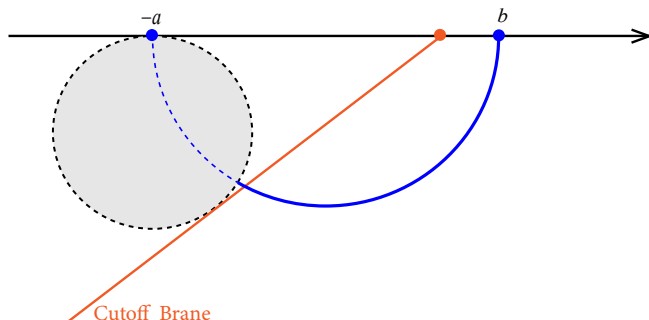

Figure 19: For the interval $\gamma = [-a, b]$ with $a > b$ in Weyl transformed CFT, the corresponding RT surface (the solid blue curve) extends behind the cutoff brane.

This is just the $\mathcal{I}(\gamma, \bar{\gamma})$ with $\gamma = [-b, b]$ in the Set-up 1, plus the additional defect term contributed from the conformal matter. Note that, when $a > b$, $\mathcal{I}(\gamma, \bar{\gamma})$ differs from the length of the bulk geodesic $\Gamma_{ab}$ connecting the boundary point $x = b$ and the point $y = -a$ on the brane (see Fig.18), plus the defect term. Specifically,

$$
\begin{aligned}
\frac{\text{Area}(\Gamma_{ab})}{4G_N} + S_{\text{defect}} &= \frac{c}{6} \log\left[\frac{b^2 + a^2 + 2ab \sin\theta_0}{a \cos\theta_0 \delta}\right] + \frac{c}{6} \log \frac{2l}{\delta_y \cos\theta_0} \\
&\neq \frac{c}{6} \log\left[\frac{(b^2 + a^2 + 2ab)(1 + \sin\theta_0)}{2a \cos\theta_0 \delta}\right] + \frac{c}{6} \log \frac{2l}{\delta_y \cos\theta_0} \\
&= \mathcal{I}(\gamma, \bar{\gamma}).
\end{aligned}
\tag{A.17}
$$

This also happens in the Set-up 1. For the interval $\gamma = [-a, b]$ with $a \neq b$ in Weyl transformed CFT, its RT surface is cutoff at the cutoff surface, rather than the cutoff brane (see Fig.19).

Now we calculate the BPE in DES model for typical configuration (see Fig.5). For $A = [0, b_2]$ and $B = [b_3, +\infty]$, the generalized islands are given by $\text{Ir}(A) = [-q, -b_2]$ and $\text{Ir}(B) = [-b_3, -q]$, where $x = -q$ is the partition point of $\text{Is}(AB) = \text{Ir}(A) \cup \text{Ir}(B)$. On the other hand, the balance point $q_2$ divides the sandwiched interval between $A$ and $B$ into $A_2 \cup B_2$ with $A_2 = [b_2, q_2]$ and $B_2 = [q_2, b_3]$. Now we solve the balance condition $\mathcal{I}(B\text{Ir}(B), A\text{Ir}(A)A_2) = \mathcal{I}(A\text{Ir}(A), B\text{Ir}(B)B_2)$. Using the generalized ALC formula and eq.(A.15), we get

$$
\begin{aligned}
\mathcal{I}(B\text{Ir}(B), A_2 A\text{Ir}(A)) &= \frac{1}{2}[\tilde{S}_{B\text{Ir}(B)} + \tilde{S}_{B_2 B\text{Ir}(B)} - \tilde{S}_{B_2}] \\
&= \frac{1}{2}(\tilde{S}_{[-\infty, -q] \cup [b_3, +\infty]} + \tilde{S}_{[-\infty, -q] \cup [q_2, +\infty]} - \tilde{S}_{[q_2, b_3]}) \\
&= \frac{c}{6} \log \frac{(b_3 + q)(q + q_2)l}{q(b_3 - q_2)\cos\theta_0 \delta_y} + \frac{c}{6}\text{arctanh}(\sin\theta_0),
\end{aligned}
\tag{A.18}
$$

$$
\begin{aligned}
\mathcal{I}(A\text{Ir}(A), B\text{Ir}(B)B_2) &= \frac{1}{2}[\tilde{S}_{A\text{Ir}(A)} + \tilde{S}_{A_2 A\text{Ir}(A)} - \tilde{S}_{A_2}] \\
&= \frac{1}{2}(\tilde{S}_{[-q, b_2]} + \tilde{S}_{[-q, q_2]} - \tilde{S}_{[b_2, q_2]}) \\
&= \frac{c}{6} \log \frac{(b_2 + q)(q_2 + q)l}{q(q_2 - b_2)\cos\theta_0 \delta_y} + \frac{c}{6}\text{arctanh}(\sin\theta_0).
\end{aligned}
$$

Then the balance condition is given by

$$\frac{b_3 + q}{b_3 - q_2} = \frac{b_2 + q}{q_2 - b_2}.$$
(A.19)

Combining the balance requirement with the minimal requirement, that is,

$$\partial_q[\mathcal{I}(B\mathrm{Ir}(B), A_2 A\mathrm{Ir}(A))] = \partial_q\left[\frac{c}{6}\log\frac{(b_3 + q)(q + q_2)l}{q(b_3 - q_2)\cos\theta_0\delta_y} + \frac{c}{6}\mathrm{arctanh}(\sin\theta_0)\right] = 0,$$
(A.20)

we arrive at

$$q = q_2 = \sqrt{b_2 b_3}.$$
(A.21)

Note that the balance point is exactly the same as the one obtained in the Weyl transformed CFT model. Finally, the BPE is given by

$$\begin{aligned}
\mathrm{BPE}(A : B) &= \mathcal{I}(B\mathrm{Ir}(B), A_2 A\mathrm{Ir}(A)) \\
&= \frac{c}{6}\log\frac{\sqrt{b_3} + \sqrt{b_2}}{\sqrt{b_3} - \sqrt{b_2}} + \frac{c}{6}\mathrm{arctanh}(\sin\theta_0) + \frac{c}{6}\log\frac{2l}{\delta_y\cos\theta_0} \\
&= \frac{\mathrm{Area}(\Sigma_{AB})}{4G_N} + S_{\mathrm{defect}},
\end{aligned}$$
(A.22)

which is exactly the reflected entropy caculated in [106]. Although the way we choose the Weyl transformation is different in these two setups, the calculation and results are essentially the same.

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
