# Peer review of "Ownerless island and partial entanglement entropy in island phases"

_SciPost Physics, doi:SciPost Phys. 15, 227 (2023)_

## Round 1 · Referee Report · Anonymous · 2023-9-14

Strengths
1. The document has been improved with addition of more exposition and set-ups.
Weaknesses
1. The motivations for studying the self-encoding property are still somewhat unclear.
Report
I thank the authors for their response and their changes. I think the current document is an improvement.
My main remaining, rather small critique would be that that the overarching motivation for considering this self-encoding property is still somewhat buried in this version. Nonetheless, based on the authors' comments (particularly their response to my point 2 in the first round), they are trying to isolate some property of quantum information that yields the island rule, and self-encoding is their working proposal. The authors mention this briefly in the new introduction, but I think they should also add a clause in the abstract that explicitly states as much.
Furthermore, as stated in the author response, the way in which self-encoding emerges in a theory of gravity is not yet clear, but it seems like this an important question to pursue (at least in the future). I would urge the authors to say that somewhere, possibly in the Discussion.
Requested changes
1. Some additional motivation of the self-encoding property (outlined in the report).
Author: Qiang Wen on 2023-09-22 [id 4003]
(in reply to Report 1 on 2023-09-14)We thank the referee for the suggestion.
The referee: My main remaining, rather small critique would be that that the overarching motivation for considering this self-encoding property is still somewhat buried in this version. Nonetheless, based on the authors' comments (particularly their response to my point 2 in the first round), they are trying to isolate some property of quantum information that yields the island rule, and self-encoding is their working proposal. The authors mention this briefly in the new introduction, but I think they should also add a clause in the abstract that explicitly states as much.
Response: The self-encoding property was proposed in the reference [49], which was written by two of the authors in this paper. It is a result of combing the island formula with our standard understanding of quantum information. We used the self-encoding property to understand the physical meaning of the two-point function of twist operators in our set-ups, and find that the two-point function should not be understand as the entanglement entropy. Then we give our basic proposal 1 to interpret the two-point function as a PEE, which is one of the cornerstone for all the calculations in this paper. The self-encoding property also helps us clarify the contribution structure to the entanglement entropy in island phases, and lead us to a new way to compute the PEE in island phases. As suggested by the referee, we will modify the abstract and some sentences in the discussion section, to clarify the role of the self-encoding property further.
The referee: Furthermore, as stated in the author response, the way in which self-encoding emerges in a theory of gravity is not yet clear, but it seems like this an important question to pursue (at least in the future). I would urge the authors to say that somewhere, possibly in the Discussion.
Response: We will mention this in the discussion section, and our group is really studying the self-encoding property of gravity by reproducing the island rules in certain set-ups from manipulating the Hilbert space.
By the way, we just replace the reference [49], which contains more discussion about why the island formula 1 is a special application of the island formula 2 in gravitational theories. Also, more discussion on why gravitational theory in island phases should self-encoded are added. Some of the statements, for example the two island formulas should be identical, are softened.

---

## Round 1 · Referee Report · Anonymous · 2023-10-13

Strengths
- Strong technical results
Weaknesses
- Claims on the applicability to non-gravitational systems which are highly non-trivial
Report
I thank the Authors for their clarifications.
I think the manuscript has improved from the previous version, especially in the clarity of some explanations. Indeed, I do not have any complaint on the technical results found and on the matching between boundary and bulk quantities.
On the other hand, the authors emphasize that the results found can also be applied in the case of non-gravitational systems, in particular the ones that display a "self encoding property" (referring to a previous paper of some of the Authors). In the manuscript, the Authors stress that "self encoding" systems comprise also non-gravitating systems, thus conjecturing an extension of the Island formula. However, this is highly non-trivial, and it is possible that self encoding systems are mostly the boundary description of a holographic (thus gravitating in the bulk) systems. This matter becomes particularly relevant when analysing holographic setups which admit an intermediate picture (between boundary and bulk), like the one of the manuscript. I think a deeper discussion of this issue (maybe with some examples of "self encoding systems" that clearly do not have a semiclassical bulk descriptions) would greatly benefit the manuscript.
I would like to stress that the comment above is not to undermine the validity of the paper, which as emphasised before I believe is technically strong, but to spark some discussion (which could very well fit into future work) in the direction of "what entanglement/encoding structures distinguished holographic (thus gravitational) theories from non gravitational ones".
Requested changes
- Possible discussion on self encoding vs holographic (gravitational) systems, if not too long. Otherwise it can be regarded as future work (see Report).

---

## Round 1 · Author Response

List of changes
The manuscript has been significally revised. See the blue colored part of the revised version and the newly added appendix.
Then main change is made for the set-up where island phase is realized. In the first version we provide a holographic Weyl transformed CFT which is non-gravitational to realize island phase (the Set-up 1). In the new version we provide two alternative set-ups, the gravitational Set-up 1 and the DES model, where gravity is coupled to half of the effective theory and the application of the Island formula can be justified.
We also give more discussion in the last section on how the BPE is related to an optimization problem.
Serveral statements, for example the self-encoding property, the Set-up 1, and the definition of the BPE, are presented in a more clear way.
Typos are fixed and several references are added.
See the reply to the referees for more details.

You are currently on this page

---

## Round 1 · List of Changes

The manuscript has been significally revised. See the blue colored part of the revised version and the newly added appendix.
Then main change is made for the set-up where island phase is realized. In the first version we provide a holographic Weyl transformed CFT which is non-gravitational to realize island phase (the Set-up 1). In the new version we provide two alternative set-ups, the gravitational Set-up 1 and the DES model, where gravity is coupled to half of the effective theory and the application of the Island formula can be justified.
We also give more discussion in the last section on how the BPE is related to an optimization problem.
Serveral statements, for example the self-encoding property, the Set-up 1, and the definition of the BPE, are presented in a more clear way.
Typos are fixed and several references are added.
See the reply to the referees for more details.

You are currently on this page

---

## Round 2 · Referee Report · Anonymous · 2023-11-13

Strengths

1- Clarified the rôle of the self-encoding property of the systems considered

Report

Dear Editor,

I thank the Authors for clarifying the rôle of the self-encoding property of the systems considered.

I now understand better their point of view, namely that, constraining the whole Hilbert space, it is possible to have non-gravitating (and truly non-holographic) systems for which an "Island Formula II" (as defined in the text) applies.

I now appreciate better that this point of view is quite original, and I agree with the Authors that it also suits laboratory implementations of systems following the "Island Formula II".

On the other hand, it is not clear whether the Hilbert space of gravitational systems is constrained in the same way as self-encoding systems. Consequently, the Authors have clarified the necessary assumptions needed for this to work, and identified future directions to resolve this issue.

Personally, I think one of the most interesting and pressing ones is (quoting the manuscript) "what are the constraints in gravitational systems that vastly reduce the Hilbert space, hence lead to the corresponding coding relation?"

Overall, I think that the manuscript has improved in clarity.

---

## Round 2 · Referee Report · Anonymous · 2023-11-18

Report

I thank the authors for addressing my remaining points in the latest version. I have nothing else to raise, so I believe the paper is ready for publication.

---

## Round 2 · List of Changes

Dear Editor

According to the referee's reports, in this version we give more clarifications on the role played by the self-encoding property in this paper. Also we give more discussion on this property and give more future directions.

We re-wrote the first paragraph in page 4, section 7.2 and the second sentence in the abstract. These parts are marked blue.

Also a few references are added

The authors

---

## Editorial Decision

published